# GATED DELTA NETWORKS:
# IMPROVING MAMBA2 WITH DELTA RULE

**Songlin Yang** [*]
MIT CSAIL
yangsl66@mit.edu

**Jan Kautz**
NVIDIA
jkautz@nvidia.com

**Ali Hatamizadeh** [*]
NVIDIA
ahatamizadeh@nvidia.com

## ABSTRACT

Linear Transformers have gained attention as efficient alternatives to standard Transformers, but their performance in retrieval and long-context tasks has been limited. To address these limitations, recent work has explored two distinct mechanisms: gating for adaptive memory control and the delta update rule for precise memory modifications. We observe that these mechanisms are complementary—gating enables rapid memory erasure while the delta rule facilitates targeted updates. Building on this insight, we introduce the gated delta rule and develop a parallel training algorithm optimized for modern hardware. Our proposed architecture, Gated DeltaNet, consistently surpasses existing models like Mamba2 and DeltaNet across multiple benchmarks, including language modeling, common-sense reasoning, in-context retrieval, length extrapolation, and long-context understanding. We further enhance performance by developing hybrid architectures that combine Gated DeltaNet layers with sliding window attention or Mamba2 layers, achieving both improved training efficiency and superior task performance. Code: https://github.com/NVlabs/GatedDeltaNet

## 1 INTRODUCTION

The Transformer architecture has significantly advanced the capabilities of Large Language Models (LLMs), showcasing exceptional performance across a wide range of tasks due to its effective attention mechanism. This mechanism excels in precise sequence modeling and leverages the parallel processing capabilities of modern GPUs during training. However, the self-attention component scales quadratically with sequence length, leading to substantial computational demands that pose challenges for both training and inference.

To mitigate these issues, researchers have explored alternatives such as linear Transformers (Katharopoulos et al., 2020a), which replace traditional softmax-based attention with kernelized dot-product-based linear attention, substantially reducing memory requirements during inference by reframing as a linear RNN with matrix-valued states. While early versions of linear Transformers underperformed in language modeling tasks compared to standard Transformers, recent enhancements—such as incorporating data-dependent gating mechanisms akin to those in LSTMs, exemplified by models like GLA (Yang et al., 2024a) and Mamba2 (Dao & Gu, 2024a)—have shown promising improvements. However, challenges persist in managing information over long sequences, particularly for in-context retrieval tasks where traditional Transformers maintain their advantage (Arora et al., 2023a; 2024a; Jelassi et al., 2024; Wen et al., 2024; Akyürek et al., 2024).

This phenomenon is not surprising: linear Transformers can be interpreted as implementing an outer-product-based key-value association memory, reminiscent of tensor product representation (Smolensky, 1990). However, the number of orthogonal key-value pairs they can store is *bounded* by the model's dimensionality. When the sequence length exceeds this dimension, "memory collisions" become inevitable, hindering exact retrieval (Schlag et al., 2021a).

Mamba2 addresses this limitation by introducing a simple gated update rule, $\mathbf{S}_t = \alpha_t \mathbf{S}_{t-1} + \boldsymbol{v}_t \boldsymbol{k}_t^\mathsf{T}$, which uniformly decays all key-value associations at each time step by a dynamic ratio, $\alpha_t \in (0, 1)$. However, this approach does not account for the varying importance of different key-value

---

[*]Equation contribution. Work done during SY's internship at NVIDIA.

associations, potentially leading to inefficient memory utilization. If the model needs to forget a specific key-value association, all key-value associations are equally forgotten, making the process less targeted and efficient.

In contrast, the linear Transformer with the delta rule (Widrow et al., 1960), known as DeltaNet (Schlag et al., 2021a; Yang et al., 2024b), selectively updates memory by (softly) replacing an old key-value pair with the incoming one in a sequential manner. This method has demonstrated impressive performance in synthetic benchmarks for in-context retrieval. However, since this process only modifies a single key-value pair at a time, the model lacks the ability to rapidly clear outdated or irrelevant information, especially during context switches where previous data needs to be erased. Consequently, DeltaNet has been found to perform moderately on real-world tasks (Yang et al., 2024b), likely due to the absence of a robust memory-clearing mechanism.

Recognizing the complementary advantages of the gated update rule and the delta rule in memory management, we propose the *gated delta rule*, a simple and intuitive mechanism that combines both approaches. This unified rule enables flexible memory control: it can promptly clear memory by setting $\alpha_t \to 0$, while selectively updating specific content without affecting other information by setting $\alpha_t \to 1$ (effectively switching to the pure delta rule).

The remaining challenge lies in implementing the gated delta rule in a hardware-efficient manner. Building upon Yang et al. (2024b)'s efficient algorithm that parallelizes the delta rule computation using the WY representation (Bischof & Loan, 1985), we carefully extend their approach to incorporate the gating terms. Our extension preserves the benefits of chunkwise parallelism (Hua et al., 2022b; Sun et al., 2023a; Yang et al., 2024a;b), enabling hardware-efficient training.

Our resulting architecture, Gated DeltaNet, consistently outperforms both Mamba2 and DeltaNet across a comprehensive suite of benchmarks, including language modeling, commonsense reasoning, in-context retrieval, length extrapolation, and long-context understanding. Building on these results, we also develop hybrid architectures that strategically combine Gated DeltaNet layers with sliding window attention or Mamba2 layers, further enhancing both training efficiency and model performance.

## 2 PRELIMINARY

### 2.1 MAMBA2: LINEAR ATTENTION WITH DECAY

It is known that the linear transformer (Katharopoulos et al., 2020b) can be formulated as the following linear recurrence when excluding normalization and query/key activations:

$$\mathbf{S}_t = \mathbf{S}_{t-1} + \boldsymbol{v}_t \boldsymbol{k}_t^\mathsf{T} \in \mathbb{R}^{d_v \times d_k}, \qquad \boldsymbol{o}_t = \mathbf{S}_t \boldsymbol{q}_t \in \mathbb{R}^{d_v}$$

where $d_k$ and $d_v$ represent the (head) dimensions for query/key and value, respectively. By expanding the recurrence, we can express it in both vector form (left) and matrix form (right) as follows:

$$\boldsymbol{o}_t = \sum_{i=1}^{t} (\boldsymbol{v}_i \boldsymbol{k}_i^\mathsf{T}) \boldsymbol{q}_t = \sum_{i=1}^{t} \boldsymbol{v}_i (\boldsymbol{k}_i^\mathsf{T} \boldsymbol{q}_t) \in \mathbb{R}^{d_v}, \qquad \mathbf{O} = (\mathbf{Q}\mathbf{K}^\mathsf{T} \odot \mathbf{M})\mathbf{V} \in \mathbb{R}^{L \times d_v}$$

where $L$ is the sequence length, and $\mathbf{M} \in \mathbb{R}^{L \times L}$ is the causal mask defined by $\mathbf{M}_{ij} = 0$ when $i < j$, and 1 otherwise.

However, this vanilla linear attention underperforms Transformers in language modeling by a large margin. To address this, it is common to add a decay term to forget historical information. Here we take Mamba2 (Dao & Gu, 2024a) as an example, which can be represented by the following linear recurrence (up to specific parameterization):

$$\mathbf{S}_t = \alpha_t \mathbf{S}_{t-1} + \boldsymbol{v}_t \boldsymbol{k}_t^\mathsf{T}, \qquad \boldsymbol{o}_t = \mathbf{S}_t \boldsymbol{q}_t$$

where $\alpha_t \in (0, 1)$ is a data-dependent scalar-valued decay term that varies with $t$. Define the cumulative decay product $\gamma_j = \prod_{i=1}^{j} \alpha_i$, and by expanding the recurrence, we can express the result in both a vector form (left) and a matrix parallel form (right):

$$\boldsymbol{o}_t = \sum_{i=1}^{t} \left( \frac{\gamma_t}{\gamma_i} \boldsymbol{v}_i \boldsymbol{k}_i^\mathsf{T} \right) \boldsymbol{q}_t = \sum_{i=1}^{t} \boldsymbol{v}_i \left( \frac{\gamma_t}{\gamma_i} \boldsymbol{k}_i^\mathsf{T} \boldsymbol{q}_t \right), \qquad \mathbf{O} = ((\mathbf{Q}\mathbf{K}^\mathsf{T}) \odot \Gamma) \mathbf{V}$$

Here, $\Gamma \in \mathbb{R}^{L \times L}$ is a decay-aware causal mask where $\Gamma_{ij} = \frac{\gamma_i}{\gamma_j}$ if $i \geq j$ and $\Gamma_{ij} = 0$ otherwise. The equivalence between these parallel and recurrent forms is also referred to as the state space duality (SSD) described in Dao & Gu (2024a). This recurrence structure appears in several other architectures including Gated RFA (Peng et al., 2021), xLSTM (Beck et al., 2024), and Gated RetNet (Sun et al., 2024b). When $\gamma_t$ is data-independent, the formulation reduces to RetNet (Sun et al., 2023a) and Lightning-Attention (Qin et al., 2024a). Furthermore, if $\gamma_t$ is extended to be matrix-valued rather than scalar-valued, efficient training algorithms remain possible when parameterized with an outer-product structure, as demonstrated by Yang et al. (2024a) and used by Yang et al. (2024a); Peng et al. (2024); Qin et al. (2024b); Zhang et al. (2024); Chou et al. (2024); He et al. (2025); Lu et al. (2025).

**Chunkwise training** However, both the recurrent and parallel forms are not ideal for efficient training (Hua et al., 2022b; Yang et al., 2024a), which motivates the use of the chunkwise parallel form (Hua et al., 2022b; Sun et al., 2023a) for hardware-efficient, linear-time training, as introduced below. To summarize, the chunkwise parallel form splits inputs and outputs into several chunks of size $C$, and computes outputs for each chunk based on the final state of the previous chunk and the query/key/value blocks of the current chunk. Following the notation of Sun et al. (2023b); Yang et al. (2024a;b), we take the query block, $\boldsymbol{q}$, as an example. We denote $\mathbf{Q}_{[t]} := \boldsymbol{q}_{tC+1:(t+1)C+1}$ as the query block for chunk $t$, and $\boldsymbol{q}_{[t]}^r := \boldsymbol{q}_{tC+r}$ as the $r$-th query within chunk $t$. The initial state of chunk $t$ is defined as $\mathbf{S}_{[t]} := \mathbf{S}_{[t]}^0 = \mathbf{S}_{[t-1]}^C$. By partially expanding the recurrence, we have

$$\mathbf{S}_{[t]}^r = \mathbf{S}_{[t]} + \sum_{i=1}^{r} \boldsymbol{v}_{[t]}^i \boldsymbol{k}_{[t]}^{i\mathsf{T}} \in \mathbb{R}^{d_v \times d_k}, \qquad \boldsymbol{o}_{[t]}^r = \mathbf{S}_{[t]}^r \boldsymbol{q}_{[t]}^r = \mathbf{S}_{[t]} \boldsymbol{q}_{[t]}^r + \sum_{i=1}^{r} \boldsymbol{v}_{[t]}^i \left( \boldsymbol{k}_{[t]}^{i\mathsf{T}} \boldsymbol{q}_{[t]}^r \right) \in \mathbb{R}^{d_v}$$

Equivalently, in matrix form:

$$\mathbf{S}_{[t+1]} = \mathbf{S}_{[t]} + \mathbf{V}_{[t]} \mathbf{K}_{[t]}^{\mathsf{T}} \in \mathbb{R}^{d_v \times d_k}, \qquad \mathbf{O}_{[t]} = \mathbf{Q}_{[t]} \mathbf{S}_{[t]}^{\mathsf{T}} + \left( \mathbf{Q}_{[t]} \mathbf{K}_{[t]}^{\mathsf{T}} \odot \mathbf{M} \right) \mathbf{V}_{[t]} \in \mathbb{R}^{C \times d_v}$$

where $\mathbf{M} \in \mathbb{R}^{C \times C}$ is the causal mask. The above equations are rich in matrix multiplications (matmuls), allowing for tensor-core-based hardware optimization. This chunkwise algorithm could be easily extended to linear attention with decay:

$$\mathbf{S}_{[t+1]} = \overrightarrow{\mathbf{S}_{[t]}} + \mathbf{V}_{[t]}^{\mathsf{T}} \overrightarrow{\mathbf{K}_{[t]}} \in \mathbb{R}^{d_v \times d_k}, \quad \mathbf{O}_{[t]} = \overleftarrow{\mathbf{Q}_{[t]}} \mathbf{S}_{[t]}^{\mathsf{T}} + \left( \mathbf{Q}_{[t]} \mathbf{K}_{[t]}^{\mathsf{T}} \odot \Gamma_{[t]} \right) \mathbf{V}_{[t]} \in \mathbb{R}^{C \times d_v} \quad (1)$$

where $(\Gamma_{[t]})_{ij} = \frac{\gamma_{[t]}^i}{\gamma_{[t]}^j}, \gamma_{[t]}^j = \prod_{j=tC+1}^{tC+j} \alpha_j$. [1] Here we use the left arrow ($\overleftarrow{\cdot}$) or the right arrow ($\overrightarrow{\cdot}$) to denote a variable decaying to the first position and the last position of each chunk, respectively,

$$\overleftarrow{\boldsymbol{q}_{[t]}^r} = \gamma_{[t]}^r \boldsymbol{q}_{[t]}^r \qquad \qquad \text{decaying each vector to the first position of chunk } t$$

$$\overrightarrow{\boldsymbol{k}_{[t]}^r} = \frac{\gamma_{[t]}^C}{\gamma_{[t]}^r} \boldsymbol{k}_{[t]}^r \qquad \qquad \text{decaying each vector to the last position of chunk } t$$

$$\overrightarrow{\mathbf{S}_{[t]}} = \gamma_{[t]}^C \mathbf{S}_{[t]} \qquad \qquad \text{decaying the state matrix over the entire chunk } t \qquad (2)$$

and likewise for other variables (e.g., $\overrightarrow{\boldsymbol{v}}$). The SSD decomposition algorithm introduced in Mamba2 is largely equivalent to this chunkwise algorithm. For a more generalized approach, Yang et al. (2024a) proposed an extended chunkwise algorithm for linear attention that incorporates fine-grained decay mechanisms.

## 2.2 DELTA NETWORKS: LINEAR ATTENTION WITH DELTA RULE

The delta update rule (Widrow et al., 1960; Schlag et al., 2021b) *dynamically* erases the value ($\boldsymbol{v}_t^{\text{old}}$) associated with the current input key ($\boldsymbol{k}_t$) and writes a new value ($\boldsymbol{v}_t^{\text{new}}$), which is a linear combination of the current input value and the old value based on the "writing strength" $\beta_t \in (0, 1)$.[2]

$$\mathbf{S}_t = \mathbf{S}_{t-1} - \underbrace{(\mathbf{S}_{t-1} \boldsymbol{k}_t)}_{\boldsymbol{v}_t^{\text{old}}} \boldsymbol{k}_t^{\mathsf{T}} + \underbrace{(\beta_t \boldsymbol{v}_t + (1 - \beta_t) \mathbf{S}_{t-1} \boldsymbol{k}_t))}_{\boldsymbol{v}_t^{\text{new}}} \boldsymbol{k}_t^{\mathsf{T}} = \mathbf{S}_{t-1} \left( \mathbf{I} - \beta_t \boldsymbol{k}_t \boldsymbol{k}_t^{\mathsf{T}} \right) + \beta_t \boldsymbol{v}_t \boldsymbol{k}_t^{\mathsf{T}}$$

---

[1] Here we slightly abuse the notation of $\gamma$ to denote the cumulative product for each chunk (starting with the first position of each chunk separately) instead of the entire sequence.

[2] It is possible to set $\beta_t \in (0, 2)$ to allow negative eigenvalue to unlock the state tracking abilities of DeltaNet (Grazzi et al., 2024; Siems et al., 2025).

As shown above, DeltaNet implements a first-order linear recurrence with generalized Householder transition matrices $(\mathbf{I} - \beta_t \boldsymbol{k}_t \boldsymbol{k}_t^\mathsf{T})$. Despite demonstrating superior associative recall and language modeling performance (Schlag et al., 2021a), DeltaNet received limited attention due to computational inefficiency until Yang et al. (2024b) introduced a hardware-efficient chunkwise training algorithm, as detailed below.

**Chunkwise parallel form.** By partially expanding the recurrence, we have

$$\mathbf{S}_{[t]}^r = \mathbf{S}_{[t]} \underbrace{\left( \prod_{i=1}^r \mathbf{I} - \beta_{[t]}^i \boldsymbol{k}_{[t]}^i \boldsymbol{k}_{[t]}^{i\mathsf{T}} \right)}_{:=\mathbf{P}_{[t]}^r} + \underbrace{\sum_{i=1}^r \left( \beta_{[t]}^i \boldsymbol{v}_{[t]}^i \boldsymbol{k}_{[t]}^{i\mathsf{T}} \prod_{j=i+1}^r \left( \mathbf{I} - \beta_{[t]}^j \boldsymbol{k}_{[t]}^j \boldsymbol{k}_{[t]}^{j\mathsf{T}} \right) \right)}_{:=\mathbf{H}_{[t]}^r} \tag{3}$$

where $\mathbf{P}_{[t]}^j$ involves cumulative products of generalized Householder matrices, which could be optimized by the classical WY representation (Bischof & Loan, 1985):

$$\mathbf{P}_{[t]}^r = \mathbf{I} - \sum_{i=1}^r \mathbf{w}_{[t]}^i \boldsymbol{k}_{[t]}^{i\mathsf{T}} \in \mathbb{R}^{d_k \times d_k} \qquad \mathbf{w}_{[t]}^r = \beta_{[t]}^r \left( \boldsymbol{k}_{[t]}^r - \sum_{i=1}^{r-1} \left( \mathbf{w}_{[t]}^i (\boldsymbol{k}_{[t]}^{i\mathsf{T}} \boldsymbol{k}_{[t]}^r) \right) \right) \in \mathbb{R}^{d_k} \tag{4}$$

Likewise, $\mathbf{H}_{[t]}^r$ could be represented as:

$$\mathbf{H}_{[t]}^r = \sum_{i=1}^r \mathbf{u}_{[t]}^i \boldsymbol{k}_{[t]}^{i\mathsf{T}} \in \mathbb{R}^{d_v \times d_k} \qquad \mathbf{u}_{[t]}^r = \beta_{[t]}^r \left( \boldsymbol{v}_{[t]}^r - \sum_{i=1}^{r-1} \left( \mathbf{u}_{[t]}^i (\boldsymbol{k}_{[t]}^{i\mathsf{T}} \boldsymbol{k}_{[t]}^r) \right) \right) \in \mathbb{R}^{d_v} \tag{5}$$

and in matrix form: $\mathbf{P}_{[t]} = \mathbf{I} - \mathbf{W}_{[t]}^\top \mathbf{K}_{[t]} \in \mathbb{R}^{d_k \times d_k}$, $\mathbf{H}_{[t]} = \mathbf{U}_{[t]}^\top \mathbf{K}_{[t]} \in \mathbb{R}^{d_v \times d_k}$. By using the UT transform (Joffrain et al., 2006), we can further write $\mathbf{W}$ and $\mathbf{U}$ in matrix form:

$$\mathbf{T}_{[t]} = \left[ \mathbf{I} + \text{strictLower} \left( \text{diag}(\beta_{[t]}) \mathbf{K}_{[t]} \mathbf{K}_{[t]}^\mathsf{T} \right) \right]^{-1} \text{diag} \left( \beta_{[t]} \right) \in \mathbb{R}^{C \times C} \tag{6}$$

$$\mathbf{W}_{[t]} = \mathbf{T}_{[t]} \mathbf{K}_{[t]} \in \mathbb{R}^{C \times d_k}, \qquad \mathbf{U}_{[t]} = \mathbf{T}_{[t]} \mathbf{V}_{[t]} \in \mathbb{R}^{C \times d_v} \tag{7}$$

Substituting these back into Eq. 3 yields a hardware-efficient chunkwise algorithm for DeltaNet that leverages matmuls, enabling tensor core based GPU optimization:

$$\mathbf{S}_{[t+1]} = \mathbf{S}_{[t]} \mathbf{P}_{[t]} + \mathbf{H}_{[t]} = \mathbf{S}_{[t]} + \left( \mathbf{U}_{[t]} - \mathbf{W}_{[t]} \mathbf{S}_{[t]}^\mathsf{T} \right)^\mathsf{T} \mathbf{K}_{[t]} \qquad \in \mathbb{R}^{d_v \times d_k} \tag{8}$$

$$\mathbf{O}_{[t]} = \mathbf{Q}_{[t]} \mathbf{S}_{[t]}^\mathsf{T} + (\mathbf{Q}_{[t]} \mathbf{K}_{[t]}^\mathsf{T} \odot \mathbf{M}) \left( \mathbf{U}_{[t]} - \mathbf{W}_{[t]} \mathbf{S}_{[t]}^\mathsf{T} \right) \qquad \in \mathbb{R}^{C \times d_v} \tag{9}$$

## 3 GATED DELTA NETWORKS

### 3.1 FORMULATION: GATED DELTA RULE

The proposed gated delta rule is simple yet effective:

$$\mathbf{S}_t = \mathbf{S}_{t-1} \left( \alpha_t (\mathbf{I} - \beta_t \boldsymbol{k}_t \boldsymbol{k}_t^\mathsf{T}) \right) + \beta_t \boldsymbol{v}_t \boldsymbol{k}_t^\mathsf{T} \tag{10}$$

where the data-dependent gating term $\alpha_t \in (0, 1)$ controls state decay. This formulation unifies the advantages of both gating mechanisms and the delta rule: the gating term enables adaptive memory management, while the delta update structure facilitates effective key-value association learning.

We present a formal analysis of the gated delta rule through the lens of the online learning framework introduced by Liu et al. (2024). In this framework, recurrent state updates emerge as *closed-form* solutions to an online learning problem, as shown in Table 1. Recent linear RNN architectures typically incorporate a regularization term in their online learning objective to prevent state divergence from previous values, thereby enabling memory retention. However, this retention mechanism becomes problematic when the state becomes saturated with information. In such cases, each state would encode a superposition of multiple information pieces, making precise retrieval challenging. To address this limitation, Mamba2 and Gated DeltaNet introduce an adaptive scaling factor $\alpha_t$ that relaxes the regularization term, allowing controlled deviations between $\mathbf{S}_t$ and $\mathbf{S}_{t-1}$. This modification enables dynamic memory management through selective forgetting, which could be useful in filtering out irrelevant information (see §3.2).

**Table 1:** Comparison of different linear RNN models and their corresponding online learning objectives using the framework from Liu et al. (2024). For convenience, we simplify Longhorn's vector-valued $\boldsymbol{\beta}$ to scalar $\beta$.

| Method | Online Learning Objective | Online Update |
|---|---|---|
| LA | $\|\mathbf{S}_t - \mathbf{S}_{t-1}\|_F^2 - 2\langle \mathbf{S}_t \boldsymbol{k}_t, \boldsymbol{v}_t\rangle$ | $\mathbf{S}_t = \mathbf{S}_{t-1} + \boldsymbol{v}_t \boldsymbol{k}_t^T$ |
| Mamba2 | $\|\mathbf{S}_t - \alpha_t \mathbf{S}_{t-1}\|_F^2 - 2\langle \mathbf{S}_t \boldsymbol{k}_t, \boldsymbol{v}_t\rangle$ | $\mathbf{S}_t = \alpha_t \mathbf{S}_{t-1} + \boldsymbol{v}_t \boldsymbol{k}_t^T$ |
| Longhorn | $\|\mathbf{S}_t - \mathbf{S}_{t-1}\|_F^2 - \beta_t \|\mathbf{S}_t \boldsymbol{k}_t - \boldsymbol{v}_t\|^2$ | $\mathbf{S}_t = \mathbf{S}_{t-1}(\mathbf{I} - \epsilon_t \boldsymbol{k}_t \boldsymbol{k}_t^T) + \epsilon_t \boldsymbol{v}_t \boldsymbol{k}_t^T, \epsilon_t = \dfrac{\beta_t}{1 + \beta_t \boldsymbol{k}_t^\top \boldsymbol{k}_t}$ |
| DeltaNet | $\|\mathbf{S}_t - \mathbf{S}_{t-1}\|_F^2 - 2\langle \mathbf{S}_t \boldsymbol{k}_t, \beta_t(\boldsymbol{v}_t - \mathbf{S}_{t-1}\boldsymbol{k}_t)\rangle$ | $\mathbf{S}_t = \mathbf{S}_{t-1}(\mathbf{I} - \beta_t \boldsymbol{k}_t \boldsymbol{k}_t^T) + \beta_t \boldsymbol{v}_t \boldsymbol{k}_t^T$ |
| Gated DeltaNet | $\|\mathbf{S}_t - \alpha_t \mathbf{S}_{t-1}\|_F^2 - 2\langle \mathbf{S}_t \boldsymbol{k}_t, \beta_t(\boldsymbol{v}_t - \alpha_t \mathbf{S}_{t-1}\boldsymbol{k}_t)\rangle$ | $\mathbf{S}_t = \mathbf{S}_{t-1}\left(\alpha_t(\mathbf{I} - \beta_t \boldsymbol{k}_t \boldsymbol{k}_t^T)\right) + \beta_t \boldsymbol{v}_t \boldsymbol{k}_t^T$ |

**Table 2:** Zero-shot performance comparison on S-NIAH benchmark suite for 1.3B models (see §4 for setups)

| | S-NIAH-1 (pass-key retrieval) | | | | S-NIAH-2 (number in haystack) | | | | S-NIAH-3 (uuid in haystack) | | |
|---|---|---|---|---|---|---|---|---|---|---|---|
| Model | 1K | 2K | 4K | 8K | 1K | 2K | 4K | 8K | 1K | 2K | 4K |
| DeltaNet | 97.4 | 96.8 | **99.0** | **98.8** | 98.4 | 45.6 | 18.6 | 14.4 | 85.2 | 47.0 | 22.4 |
| Mamba2 | **99.2** | **98.8** | 65.4 | 30.4 | 99.4 | 98.8 | 56.2 | 17.0 | 64.4 | 47.6 | 4.6 |
| **Gated DeltaNet** | 98.4 | 88.4 | 91.4 | 91.8 | **100.0** | **99.8** | **92.2** | **29.6** | **86.6** | **84.2** | **27.6** |

On the other hand, Linear Attention (LA) and Mamba2 use a simple negative inner-product loss $-\langle \mathbf{S}_t \boldsymbol{k}_t, \boldsymbol{v}_t\rangle$, while Longhorn (Liu et al., 2024) uses a more expressive online regression objective $\|\mathbf{S}_t \boldsymbol{k}_t - \boldsymbol{v}_t\|^2$ for better modeling of key-value associations. The resulting Longhorn's update rule closely resembles the delta update rule, [3] suggesting the superiority of the (gated) delta rule over Mamba2 in in-context associative recall.

From the perspective of fast weight programming (Irie et al., 2022a) and test-time training (Sun et al., 2024a) and regression (Wang et al., 2025), the hidden state $\mathbf{S}$ can be interpreted as a (fast) weight matrix, with the delta rule optimizing the online regression objective $\mathcal{L}(\mathbf{S}_t) = \frac{1}{2}\|\mathbf{S}_t \boldsymbol{k}_t - \boldsymbol{v}_t\|^2$ via *test-time* stochastic gradient descent (SGD):

$$\mathbf{S}_{t+1} = \mathbf{S}_t - \beta_t \nabla \mathcal{L}(\mathbf{S}_t) = \mathbf{S}_t - \beta_t(\mathbf{S}_t \boldsymbol{k}_t - \boldsymbol{v}_t)\boldsymbol{k}_t^\top = \mathbf{S}_t\left(\mathbf{I} - \beta_t \boldsymbol{k}_t \boldsymbol{k}_t^\top\right) + \beta_t \boldsymbol{v}_t \boldsymbol{k}_t^\top$$

where $\beta_t$ represents the (adaptive) learning rate. From this perspective, the gated delta rule can be viewed as incorporating an adaptive weight decay term $\alpha_t$ into the SGD update, a technique widely used in deep learning (Krogh & Hertz, 1991; Andriushchenko et al., 2023). Concurrently, Titans (Behrouz et al., 2024) demonstrated the effectiveness of incorporating weight decay mechanisms in RNN test-time SGD updates.

### 3.2 CASE STUDY: SINGLE NEEDLE IN A HAYSTACK (S-NIAH)

To better understand the complementary strength between the delta rule and the gated rule, we present a case study on the Single Needle-In-A-Haystack (S-NIAH) benchmark suite from RULER (Hsieh et al., 2024), where a key-value pair acts as a needle in the haystack (context) and the model must recall the value when given the key. Table 2 presents the results and we draw three main observations:

**Decay hurts memory retention.** In the simplest S-NIAH-1 setting with repeated synthetic context, models memorize minimal information, testing long-term retention. DeltaNet achieves near-perfect performance across all sequence lengths. Mamba2 degrades significantly beyond 2K sequences since it decays historical information too quickly, while Gated DeltaNet's degradation is less severe thanks to the use of delta rule.

**Gating facilitates filtering.** In S-NIAH-2/3 with real-world-essay context, models store all potentially relevant information, testing efficient memory management. With fixed state size, lack of clearance causes memory collision—information becomes superimposed and indistinguishable. DeltaNet's performance drops significantly at longer sequences due to poor memory clearance. Mamba2 and Gated DeltaNet maintain better performance through gating mechanisms that filter irrelevant information.

---

[3]The theoretical distinction lies in the optimization approach: Longhorn uses implicit online learning (Kulis & Bartlett, 2010) to derive closed-form globally optimal updates, while DeltaNet optimizes the same objective through one-step explicit gradient descent, as noted by Liu et al. (2024).

**Delta rule helps memorization.** In S-NIAH-3, values change from numbers to UUIDs, testing complex pattern memorization. Mamba2's performance drops quickly, while Gated DeltaNet performs better, verifying that the delta rule indeed has better memorization ability.

### 3.3 ALGORITHM: HARDWARE-EFFICIENT CHUNKWISE TRAINING

In this subsection, we derive a hardware-efficient chunkwise algorithm for training Gated DeltaNet. By partially expanding the recurrence in Eq. 10, we have

$$\mathbf{S}_{[t]}^r = \mathbf{S}_{[t]} \underbrace{\left( \prod_{i=1}^r \alpha_{[t]}^i \left( \mathbf{I} - \beta_{[t]}^i \mathbf{k}_{[t]}^i \mathbf{k}_{[t]}^{i\mathsf{T}} \right) \right)}_{:=\mathbf{F}_{[t]}^r} + \underbrace{\sum_{i=1}^r \left( \beta_{[t]}^i \mathbf{v}_{[t]}^i \mathbf{k}_{[t]}^{i\mathsf{T}} \prod_{j=i+1}^r \alpha_{[t]}^j \left( \mathbf{I} - \beta_{[t]}^j \mathbf{k}_{[t]}^j \mathbf{k}_{[t]}^{j\mathsf{T}} \right) \right)}_{:=\mathbf{G}_{[t]}^r}$$

It is easy to see that $\mathbf{F}_{[t]}^r = \gamma_{[t]}^r \mathrm{P}_{[t]}^{\mathrm{r}} = \overleftarrow{\mathrm{P}_{[t]}^{\mathrm{r}}}$. As for $\mathbf{G}_{[t]}^r$, we adapt Eq. 5 as follows,

$$\mathbf{G}_{[t]}^r = \sum_{i=1}^r \frac{\gamma_{[t]}^r}{\gamma_{[t]}^i} \tilde{\mathbf{u}}_{[t]}^i \mathbf{k}_{[t]}^{i\mathsf{T}} \in \mathbb{R}^{d_v \times d_k} \qquad \tilde{\mathbf{u}}_{[t]}^r = \beta_{[t]}^r \left( \mathbf{v}_{[t]}^r - \sum_{i=1}^{r-1} \left( \tilde{\mathbf{u}}_{[t]}^i \left( \frac{\gamma_{[t]}^r}{\gamma_{[t]}^i} \mathbf{k}_{[t]}^{i\mathsf{T}} \mathbf{k}_{[t]}^r \right) \right) \right) \in \mathbb{R}^{d_v}$$

(see §A for a proof). By UT transform, we have the matrix form:

$$\widetilde{\mathbf{U}_{[t]}} = \left[ \mathbf{I} + \mathrm{strictLower} \left( \mathrm{diag} \left( \beta_{[t]} \right) \left( \Gamma_{[t]} \odot \mathbf{K}_{[t]} \mathbf{K}_{[t]}^{\mathsf{T}} \right) \right) \right]^{-1} \mathrm{diag} \left( \beta_{[t]} \right) \mathbf{V}_{[t]} \qquad \in \mathbb{R}^{C \times d_v}$$

Similar to how Mamba2 extends linear attention (Eq. 1), we can adapt DeltaNet's chunkwise algorithm (Eq. 8-9) for Gated DeltaNet to enable hardware-efficient training as follows:

$$\mathbf{S}_{[t+1]} = \overrightarrow{\mathbf{S}_{[t]}} + \left( \widetilde{\mathbf{U}_{[t]}} - \overleftarrow{\mathbf{W}_{[t]}} \mathbf{S}_{[t]}^{\mathsf{T}} \right)^{\mathsf{T}} \overrightarrow{\mathbf{K}_{[t]}} \qquad \in \mathbb{R}^{d_v \times d_k}$$

$$\mathbf{O}_{[t]} = \overleftarrow{\mathbf{Q}_{[t]}} \mathbf{S}_{[t]}^{\mathsf{T}} + \left( \mathbf{Q}_{[t]} \mathbf{K}_{[t]}^{\mathsf{T}} \odot \mathbf{M} \right) \left( \widetilde{\mathbf{U}_{[t]}} - \overleftarrow{\mathbf{W}_{[t]}} \mathbf{S}_{[t]}^{\mathsf{T}} \right) \qquad \in \mathbb{R}^{C \times d_v}$$

where $\overleftarrow{\mathbf{q}_{[t]}^r} = \gamma_{[t]}^r \mathbf{q}_{[t]}^r$, $\overleftarrow{\mathbf{w}_{[t]}^r} = \gamma_{[t]}^r \mathbf{w}_{[t]}^r$, $\overrightarrow{\mathbf{k}_{[t]}^r} = \frac{\gamma_{[t]}^C}{\gamma_{[t]}^r} \mathbf{k}_{[t]}^r$, and $\overrightarrow{\mathbf{S}_{[t]}} = \gamma_{[t]}^C \mathbf{S}_{[t]}$ like we defined in Eq. 2.

### 3.4 GATED DELTA NETWORKS AND HYBRID MODELS

**Token mixer block.** The basic Gated DeltaNet follows Llama's macro architecture, stacking token mixer layers with SwiGLU MLP layers, but replaces self-attention with gated delta rule token mixing. Fig. 1 (right) shows its block design. For the gated delta rule (Eq. 10), queries, keys and values $\{\mathbf{q}, \mathbf{k}, \mathbf{v}\}$ are generated through linear projection, short convolution and SiLU, with L2 normalization applied to $\mathbf{q}, \mathbf{k}$ for training stability. $\alpha, \beta$ use linear projection only.[4] Following Sun et al. (2023a), the output is processed through normalization and gating before applying output projection.

**Hybrid models.** Linear transformers have limitations in modeling local shifts and comparisons, and their fixed state size makes it hard for retrieval tasks (Arora et al., 2024a). Following recent hybrid architectures like Griffin (De et al., 2024) and Samba (Ren et al., 2024), we combine linear recurrent layers with sliding window attention (SWA), resulting in GatedDeltaNet-H1. We also stack Mamba2, GatedDeltaNet and SWA, resulting in GatedDeltaNet-H2.

## 4 EXPERIMENTS

**Setup** Our experiments encompass a comprehensive comparison of recent state-of-the-art architectures, including pure Transformer models, RNN-based approaches, and hybrid architectures. We evaluate against the following baselines: RetNet (Sun et al., 2023a), HGRN2 (Qin et al., 2024b), Mamba (Gu & Dao, 2023), Mamba2 (Dao & Gu, 2024b), Samba (Ren et al., 2024), and DeltaNet (Yang et al., 2024b). For fair comparison, all models are trained under identical conditions with 1.3B parameters on 100B tokens sampled from the FineWeb-Edu dataset (Penedo et al., 2024). We use the AdamW optimizer with a peak learning rate of 4e-4, weight decay of 0.1, and gradient clipping of 1.0. The learning rate follows a cosine annealing schedule with a 1B token warm-up period and batch size of 0.5M tokens. All models employ the Llama2 tokenizer with a vocabulary size of 32,000. For sequence modeling, we set the training length to 4K tokens, with Samba and our hybrid models using a sliding window size of 2K. See § B.1 for evaluation settings and § B.2 for ablation studies.

---

[4] We use Mamba2's parameterization for $\alpha$ but omit it for brevity.

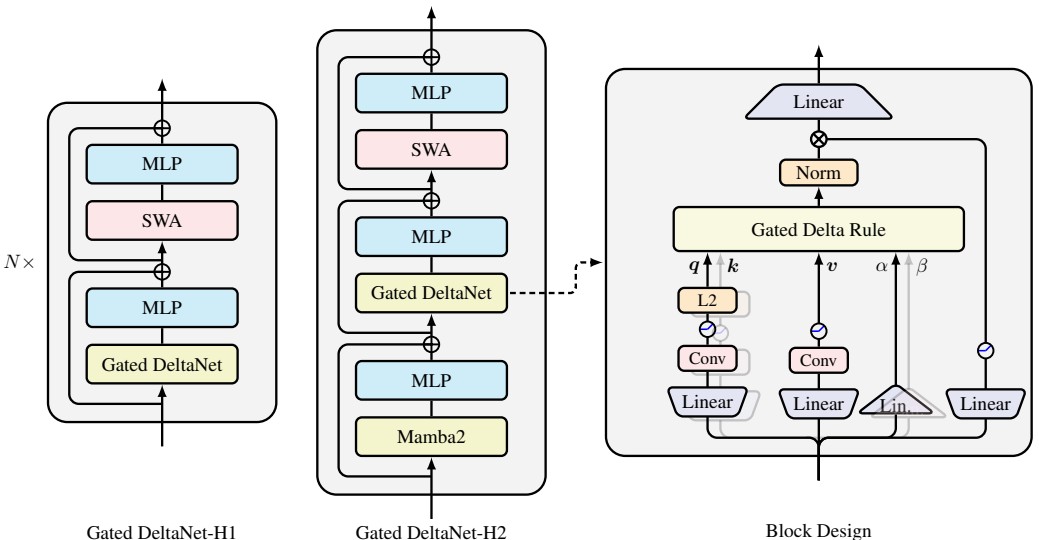

**Figure 1:** Visualization of the (hybrid) architecture and block design of Gated DeltaNet models. Gated DeltaNet-H1 and H2 use Gated DeltaNet + SWA and Mamba2 + Gated DeltaNet + SWA patterns, respectively. In the block design, query/key paths consist of linear proj., shortconv., SiLU and L2 norm; value path includes linear proj., shortconv. and SiLU; alpha/beta use linear proj.; and output gate applies linear proj. with SiLU.

| Model | Wiki. ppl ↓ | LMB. ppl ↓ | LMB. acc ↑ | PIQA acc ↑ | Hella. acc_n ↑ | Wino. acc ↑ | ARC-e acc ↑ | ARC-c acc_n ↑ | SIQA acc ↑ | BoolQ acc ↑ | Avg. |
|---|---|---|---|---|---|---|---|---|---|---|---|
| *Recurrent models* | | | | | | | | | | | |
| RetNet | 19.08 | 17.27 | 40.52 | 70.07 | 49.16 | 54.14 | 67.34 | 33.78 | **40.78** | 60.39 | 52.02 |
| HGRN2 | 19.10 | 17.69 | 39.54 | 70.45 | 49.53 | 52.80 | 69.40 | 35.32 | 40.63 | 56.66 | 51.79 |
| Mamba | 17.92 | 15.06 | 43.98 | 71.32 | 52.91 | 52.95 | 69.52 | 35.40 | 37.76 | **61.13** | 53.12 |
| Mamba2 | 16.56 | 12.56 | 45.66 | 71.87 | 55.67 | 55.24 | 72.47 | 37.88 | 40.20 | 60.13 | 54.89 |
| DeltaNet | 17.71 | 16.88 | 42.46 | 70.72 | 50.93 | 53.35 | 68.47 | 35.66 | 40.22 | 55.29 | 52.14 |
| Gated DeltaNet | **16.42** | **12.17** | 46.65 | 72.25 | 55.76 | 57.45 | 71.21 | 38.39 | 40.63 | 60.24 | 55.32 |
| *Attention or hybrid models* | | | | | | | | | | | |
| Transformer++ | 18.53 | 18.32 | 42.60 | 70.02 | 50.23 | 53.51 | 68.83 | 35.10 | 40.66 | 57.09 | 52.25 |
| Samba | 16.13 | 13.29 | 44.94 | 70.94 | 53.42 | 55.56 | 68.81 | 36.17 | 39.96 | 62.11 | 54.00 |
| Gated DeltaNet-H1 | 16.07 | **12.12** | 47.73 | **72.57** | 56.53 | **58.40** | 71.75 | 40.10 | 41.40 | 63.21 | 56.40 |
| Gated DeltaNet-H2 | **15.91** | 12.55 | 48.76 | 72.19 | 56.88 | 57.77 | 71.33 | 39.07 | 41.91 | 61.55 | 56.18 |

**Table 3:** Performance comparison on language modeling and zero-shot common-sense reasoning.

**Common-sense reasoning** In Table 3, we present the language modeling perplexity and **zero-shot** accuracy on common-sense reasoning benchmarks for models with 400M and 1.3B parameters. Gated DeltaNet consistently outperforms other linear models, including RetNet, HGRN2, Mamba, Mamba2, and DeltaNet, at both scales. As expected, the hybrid variant further enhances performance.

| Models | SWDE | SQD | FDA | TQA | NQ | Drop | Avg |
|---|---|---|---|---|---|---|---|
| *Recurrent models* | | | | | | | |
| RetNet | 14.0 | 28.5 | 7.0 | 54.4 | 16.2 | 17.3 | 22.9 |
| HGRN2 | 8.3 | 25.3 | 4.8 | 51.2 | 14.2 | 16.9 | 20.1 |
| Mamba | 9.8 | 25.8 | 3.7 | 54.3 | 14.9 | 17.4 | 21.0 |
| Mamba2 | 19.1 | 33.6 | **25.3** | **61.0** | **20.8** | 19.2 | 29.8 |
| DeltaNet | 17.9 | 30.9 | 18.4 | 53.9 | 17.3 | 18.6 | 26.2 |
| Gated DeltaNet | **25.4** | **34.8** | 23.7 | 60.0 | 20.0 | **19.8** | **30.6** |
| *Attention or hybrid models* | | | | | | | |
| Transformer++ | 29.5 | 38.0 | **52.2** | 58.3 | 22.5 | 21.6 | 37.0 |
| Samba | 33.0 | 39.2 | 50.5 | 57.7 | 23.5 | 20.2 | 37.3 |
| Gated DeltaNet-H1 | 35.6 | 39.7 | 52.0 | 60.1 | 24.6 | 22.2 | 39.0 |
| Gated DeltaNet-H2 | **38.2** | **40.4** | 50.7 | **63.3** | 24.8 | 23.3 | **40.1** |

**Table 4:** Accuracy on recall-world retrieval tasks with input truncated to 2K tokens. SQD: SQUADE. TQA: Trivial QA.

**In-context retrieval on real-world data** Table 4 presents results on real-world recall-intensive tasks used by Arora et al. (2024b). As expected, linear recurrent models show a significant performance gap compared to Transformers, while hybrid models combining linear recurrence and attention outperform pure attention models in retrieval tasks.

For pure recurrent models, despite DeltaNet's superior performance on synthetic in-context retrieval tasks (Yang et al., 2024b), its real-world retrieval performance lags behind Mamba2, consistent with our observations in S-NIAH-2 and S-NIAH-3 (Table 2). Gated DeltaNet outperforms both DeltaNet and Mamba2 thanks to its gated delta rule, though the improvement margin is smaller than in Table 2. We attribute this reduced performance gap to instruction-unaligned small language models being prone to repetition errors, which are the primary source of errors in these tasks (cf. Arora et al. (2024b, Appendix E)). Since this issue is largely independent of the update rule choice, the performance differences between models are less pronounced compared to Table 2.

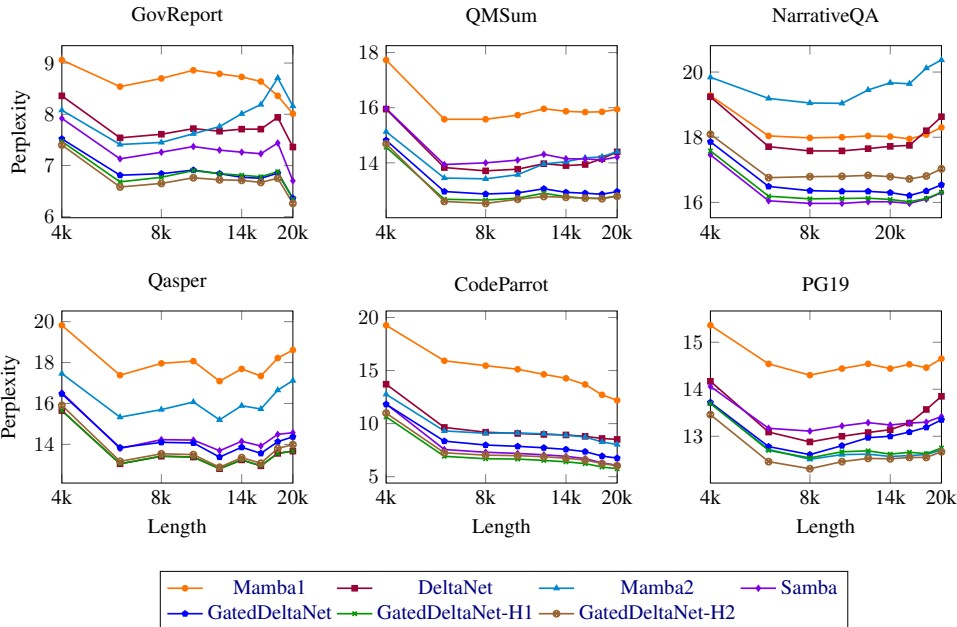

**Figure 2:** Length extrapolation on six long benchmarks.

**Length extrapolation on long sequences.** As shown in Fig.2, we evaluate the models' capacity to extrapolate to sequences of up to 20K tokens across six long-context benchmarks. Gated DeltaNet achieves the lowest overall perplexity across tasks among RNN models. While we observe mixed results in length extrapolation, Gated DeltaNet exhibits relatively more robust performance, suggesting better memory management. The hybrid models further improve upon this by leveraging attention for local context modeling, which reduces the memory management burden on their recurrent components. Future work will explore these models' capabilities on even longer sequences.

**Long context understanding** As demonstrated in Table 5, we evaluated the models' performance on LongBench (Bai et al., 2023). In recurrent models, Gated DeltaNet shows consistent advantages, especially in single-doc QA, few-shot in-context learning, and Code tasks, demonstrating its superior capabilities in retrieval, in-context learning, and state tracking, respectively.

**Throughput Comparison.** The training throughput comparison across different models is presented in Fig. 3. As our analysis shows, the proposed gated delta rule introduces only marginal overhead compared to the original delta rule, with Gated DeltaNet achieving essentially the same throughput as DeltaNet. Both are slightly slower than Mamba2 (2-3K tokens/sec) due to their more expressive transition matrices.

The Transformer++ achieves the best performance in the 2K context window domain, thanks to the highly optimized Flash-Attention-2 kernel (Dao, 2023). Consequently, hybrid approaches combining 2K window-size SWA attention with other token mixers demonstrate higher throughput than standalone mixers: Samba outperforms Mamba, while Gated DeltaNet-H1 and -H2 outperform Gated DeltaNet. Notably, Gated DeltaNet-H1 maintains compelling training throughput across all sequence lengths, even on short sequences.

| Model | Single-Doc QA | | | Multi-Doc QA | | | Summarization | | | Few-shot | | | Code | | Avg |
|---|---|---|---|---|---|---|---|---|---|---|---|---|---|---|---|
| | NQA | QQA | MFQ | HQA | 2WM | Mus | GvR | QMS | MNs | TRC | TQA | SSM | LCC | RBP | |
| *Recurrent models* | | | | | | | | | | | | | | | |
| RetNet | 12.1 | 10.7 | 19.1 | 10.7 | **18.0** | 5.8 | 4.8 | 15.8 | 7.9 | 19.0 | 18.0 | 12.8 | 14.1 | 17.9 | 13.2 |
| HGRN2 | 10.7 | 12.1 | 19.1 | 11.3 | 15.7 | 6.0 | 5.2 | 15.1 | 9.2 | 16.0 | 15.8 | 10.3 | 18.6 | 20.8 | 13.5 |
| Mamba | 13.0 | 10.1 | 20.4 | 10.1 | 16.7 | 6.0 | 7.2 | 15.9 | 8.4 | 23.1 | 21.9 | 11.2 | 17.9 | 19.0 | 14.6 |
| DeltaNet | 12.9 | 10.8 | 21.5 | 10.9 | 13.2 | 5.1 | 6.5 | 13.5 | 7.2 | 15.5 | 23.3 | 11.6 | 17.6 | 20.3 | 13.6 |
| Mamba2 | 11.1 | 11.3 | 18.6 | 11.8 | 15.1 | 6.7 | 6.7 | 14.5 | 7.4 | 13.0 | 23.6 | 8.4 | 17.9 | 20.6 | 13.5 |
| **Gated DeltaNet** | **14.1** | **14.0** | **23.3** | **13.7** | 14.4 | 5.8 | **7.5** | **16.4** | 7.9 | **30.0** | 22.4 | **23.0** | **18.7** | **22.1** | **16.6** |
| *Attention or hyrbid models* | | | | | | | | | | | | | | | |
| Transformer++ | 11.8 | 9.3 | 10.0 | 10.9 | 4.2 | 6.1 | 7.4 | 15.8 | 6.6 | 16.9 | 13.5 | 3.9 | 17.2 | 18.7 | 11.0 |
| Samba | 12.5 | 12.9 | 25.4 | 11.2 | 19.7 | 6.8 | 9.1 | 15.7 | 11.0 | 20.0 | 22.7 | 22.8 | 18.1 | 21.1 | 15.9 |
| **Gated DeltaNet-H1** | **14.5** | 12.3 | **26.6** | 12.6 | **23.6** | 6.1 | 9.1 | 16.1 | 12.8 | 33.5 | 23.9 | 26.8 | 15.5 | 19.2 | 17.8 |
| **Gated DeltaNet-H2** | 12.7 | **13.0** | **27.1** | **12.7** | 20.6 | 7.5 | 10.4 | 16.2 | 13.0 | 40.5 | 22.7 | 27.9 | 19.9 | 22.1 | 18.4 |

**Table 5:** Accuracy on 14 tasks from LongBench (Bai et al., 2023): Narrative QA, QasperQA, MultiField QA, HotpotQA, 2WikiMulti QA, Musique, GovReport, QMSum, MultiNews, TRec, Trivia QA, SamSum, LCC, and RepoBench-P by order.

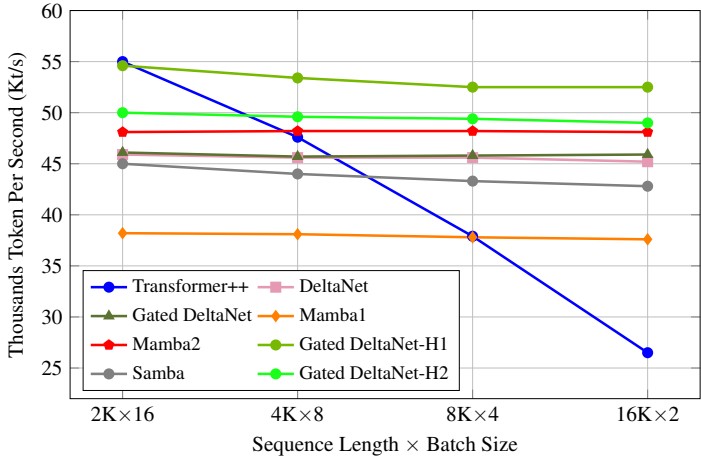

**Figure 3:** Training throughput comparison of 1.3B models on a single H100 GPU.

# 5 RELATED WORK

**Gated linear RNN.** Large linear recurrent language models have attracted significant attention due to their training and inference efficiency. The field of linear RNNs has rapidly evolved from using data-independent decay mechanisms, as exemplified by models like S4 (Gu et al., 2022), S5 (Smith et al., 2023), LRU (Orvieto et al., 2023), RWKV4/5 (Peng et al., 2023), and RetNet (Sun et al., 2023a), to incorporating data-dependent decay mechanisms in more recent architectures such as HGRN1/2 (Qin et al., 2024b; 2023b), Mamba1/2 (Gu & Dao, 2023; Dao & Gu, 2024a), RWKV6 (Peng et al., 2024), GSA (Zhang et al., 2024). This transition stems from the proven advantages of gating/forgetting mechanisms (termed selective mechanisms in Mamba)—a classical concept originating in the gated RNN literature (Gers et al., 2000) whose significance has been consistently reaffirmed (Greff et al., 2015; van der Westhuizen & Lasenby, 2018; Qin et al., 2024b; 2023b; Gu & Dao, 2023).

Modern forget gates differ from traditional designs like those in LSTM by removing the dependency on the previous hidden state, relying solely on input data. This modification enables efficient parallelism across sequence lengths (Martin & Cundy, 2018; Qin et al., 2023b). The absence of a forget gate has been a notable limitation in DeltaNet, and our gated extension addresses this gap in a natural, effective, and hardware-efficient way. We also note a recent concurrent work RWKV-7 [5] using a similar idea, but with a more relaxable formalism using diagonal-plus-low-rank transitions:

---

[5]https://github.com/BlinkDL/RWKV-LM/tree/main/RWKV-v7

$\mathbf{S}_t = \mathbf{S}_{t-1}(\mathrm{diag}(\mathbf{d}_t) - \mathbf{a}_t \mathbf{b}_t^\top) + \boldsymbol{v}_t \boldsymbol{k}_t^\top$ where $\mathbf{d}_t, \mathbf{a}_t, \mathbf{b}_t \in \mathbb{R}^{d_k}$. The chunkwise algorithm could be similarly adapted to this case, as implemented in Flash Linear Attention (Yang & Zhang, 2024). [6]

**Delta rule.**  The delta learning rule demonstrates superior memory capacity compared to Hebbian learning (Gardner, 1988; Prados & Kak, 1989), an advantage DeltaNet leverages while linear transformers rely on Hebbian-like rules. This memory capacity advantage is evident in synthetic in-context learning tasks and extends to language modeling (Irie et al., 2021; Yang et al., 2024b), reinforcement learning (Irie et al., 2022b), and image generation (Irie & Schmidhuber, 2023). Yang et al. (2024b) parallelized delta rule computation and demonstrated how DeltaNet's data-dependent identity-plus-low-rank structure $(\mathbf{I} - \beta_t \boldsymbol{k}_t \boldsymbol{k}_t^\intercal)$ offers greater flexibility than Mamba2's data-dependent diagonal matrices $(\alpha_t \mathbf{I})$. This structural advantage could enable complex reasoning, including regular language recognition (Fan et al., 2024; Grazzi et al., 2024) and state-tracking beyond $\mathrm{TC}^0$ complexity (Merrill et al., 2024)—crucial for coding and reasoning applications.

Despite these significant advantages, the delta rule faces theoretical limitations (Irie et al., 2023) and shows only moderate performance on real-world datasets (Yang et al., 2024b), suggesting room for improvement. Previous attempts to enhance expressiveness through nonlinear recurrence (Irie et al., 2021; 2022b) addressed some limitations but sacrificed training parallelism, creating a performance-efficiency tradeoff. Recent work proposes some enhancements without compromising parallelism for better state tracking performance, including using negative eigenvalues (Grazzi et al., 2024) and multiple products of householder transition matrices (Siems et al., 2025) which enable high-rank transformations. These methods could be applied to Gated DeltaNet seamlessly.

From a (online) learning objective perspective, alternative formulations could further extend expressiveness: nonlinear regression $(\mathcal{L}(\mathbf{S}_t) = \frac{1}{2}||f_{\mathbf{S}_t}(\boldsymbol{k}_t) - \boldsymbol{v}_t||^2)$ as in TTT (Sun et al., 2024a) and Titans (Behrouz et al., 2024), where $f_{\mathbf{S}}$ is a nonlinear function parameterized by $\mathbf{S}$; or regression considering the entire history $(\mathcal{L}(\mathbf{S}_t) = \frac{1}{2}\sum_{i=1}^t ||\mathbf{S}_t \boldsymbol{k}_i - \boldsymbol{v}_i||^2)$ as in Mesa layer (von Oswald et al., 2024)—analogous to the difference between Least Mean Square and Recursive Least Square algorithms. However, these more expressive variants introduce nonlinear recurrence and require workarounds, such as performing nonlinear updates only after processing entire chunks (as in TTT and Titans); or approximating nonlinear recurrence methods like Lim et al. (2024); Gonzalez et al. (2024); Schöne et al. (2025).

**Hybrid models.**  In this work, we explore interleaving hybrid attention layers across layers, which is commonly used such as in MiniMax-01 (MiniMax et al., 2025) and Hybrid Mamba2-Attention (Waleffe et al., 2024). It is also interesting to investigate hybrid linear/softmax attention within a single layer (Hua et al., 2022a; Zancato et al., 2024; Munkhdalai et al., 2024; Nunez et al., 2024; Dong et al., 2025; Zhang et al., 2025).

## 6  CONCLUSION

In this work, we introduced Gated DeltaNet, which enables better key-value association learning compared to Mamba2 and more adaptive memory clearance than DeltaNet, leading to consistently better empirical results across various tasks. We extended the parallel algorithm from Yang et al. (2024b) to enable hardware-efficient training of Gated DeltaNet. Our hybrid Gated DeltaNet model achieves even higher training throughput and overall performance, making it well-suited for practical deployment.

## ACKNOWLEDGMENT

We thank Yu Zhang for assistance with figure creation and model evaluation; Kazuki Irie for providing valuable feedback on the draft; Simeng Sun and Zhixuan Lin for insightful discussions on long-sequence task evaluation settings; and Eric Alcaide and Volodymyr Kyrylov for their helpful discussions on the online learning perspective of DeltaNet.

---

[6] `https://github.com/fla-org/flash-linear-attention/tree/main/fla/ops/generalized_delta_rule`.

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

## A  EXTENDED WY REPRESENTATION FOR GATED DELTA RULE

To reduce notation clutter, we only consider the first chunk here.

For $\mathbf{S}_t$, the extended WY representation is

$$\mathbf{S}_t = \sum_{i=1}^{t} \frac{\gamma_t}{\gamma_i} \mathbf{u}_i \boldsymbol{k}_i^\mathsf{T}, \qquad \mathbf{u}_t = \beta_t \left( \boldsymbol{v}_t - \sum_{i=1}^{t-1} \frac{\gamma_t}{\gamma_i} \mathbf{u}_i \boldsymbol{k}_i^T \boldsymbol{k}_t \right)$$

We proof this by mathmetical induction.

*Proof.*

$$\mathbf{S}_{t+1} = \mathbf{S}_t \left( \alpha_{t+1}(\mathbf{I} - \beta_{t+1}\boldsymbol{k}_{t+1}\boldsymbol{k}_{t+1}^\mathsf{T}) \right) + \beta_{t+1}\boldsymbol{v}_{t+1}\boldsymbol{k}_{t+1}^\mathsf{T}$$

$$= \alpha_{t+1}(\sum_{i=1}^{t} \frac{\gamma_t}{\gamma_i} \mathbf{u}_i \boldsymbol{k}_i^\mathsf{T}) - \alpha_{t+1}\beta_{t+1}(\sum_{i=1}^{t} \frac{\gamma_t}{\gamma_i} \mathbf{u}_i \boldsymbol{k}_i^\mathsf{T} \boldsymbol{k}_i \boldsymbol{k}_{t+1}^\mathsf{T}) + \beta_{t+1}\boldsymbol{v}_{t+1}\boldsymbol{k}_{t+1}^\mathsf{T}$$

$$= \sum_{i=1}^{t} \frac{\gamma_{t+1}}{\gamma_i} \mathbf{u}_i \boldsymbol{k}_i^\mathsf{T} + \underbrace{\beta_{t+1} \left( \boldsymbol{v}_{t+1} - \sum_{i=1}^{t} \frac{\gamma_{t+1}}{\gamma_i} \mathbf{u}_i \boldsymbol{k}_i^T \boldsymbol{k}_{t+1} \right)}_{\mathbf{u}_{t+1}} \boldsymbol{k}_{t+1}^\mathsf{T}$$

$$= \sum_{i=1}^{t} \frac{\gamma_{t+1}}{\gamma_i} \mathbf{u}_i \boldsymbol{k}_i^\mathsf{T} + \underbrace{\frac{\gamma_{t+1}}{\gamma_{t+1}}}_{1} \mathbf{u}_{t+1} \boldsymbol{k}_{t+1}^\mathsf{T}$$

$$= \sum_{i=1}^{t+1} \frac{\gamma_{t+1}}{\gamma_i} \mathbf{u}_i \boldsymbol{k}_i^\mathsf{T}$$

$\square$

## B  EXPERIMENT CONTUNUED

### B.1  EVALUATION

**Commonsense reasoning**    Following Gu & Dao (2023), we evaluate our model on multiple commonsense reasoning benchmarks: PIQA (Bisk et al., 2020), HellaSwag (Hella.; Zellers et al., 2019), WinoGrande (Wino.; Sakaguchi et al., 2020), ARC-easy (ARC-e) and ARC-challenge (ARC-c) (Clark et al., 2018), SIQA (Sap et al., 2019), BoolQ (Clark et al., 2019), Wikitext (Wiki.; Merity et al., 2017), and LAMBADA (LMB.; Paperno et al., 2016).

**In-context retrieval**    Our evaluation comprises both synthetic and real-world tasks. For synthetic tasks, we utilize the Needle-In-A-Haystack Single (NIAH-S) benchmark suite from RULER (Hsieh et al., 2024), which includes three increasingly complex tasks: S-NIAH-1 (passkey retrieval), S-NIAH-2 (numerical needle in haystack), and S-NIAH-3 (word-based needle in haystack). For real-world tasks, following Arora et al. (2024b), we evaluate on diverse datasets: SWDE (Lockard et al., 2019) for structured HTML relation extraction, FDA (Arora et al., 2023b) for PDF key-value retrieval, and several question-answering datasets including SQuAD (Rajpurkar et al., 2018), TriviaQA (Joshi et al., 2017a), Drop (Dua et al., 2019), and NQ (Kwiatkowski et al., 2019). Since our pretrained models lack instruction tuning, we employ the Cloze Completion Formatting prompts provided by Arora et al. (2024b), which better align with our models' next-word-prediction training objective.

**Long context understanding**    We evaluate on 14 tasks from Longbench (Bai et al., 2023), encompassing: narrative comprehension (Narrative QA (Kočiský et al., 2018)), scientific understanding (QasperQA (Dasigi et al., 2021)), multi-hop reasoning (MultiField QA, HotpotQA (Yang et al., 2018), 2WikiMulti QA (Ho et al., 2020), Musique (Trivedi et al., 2022)), document summarization (GovReport (Huang et al., 2021), QMSum (Zhong et al., 2021), MultiNews (Fabbri et al., 2019)), and various specialized tasks (TRec (Li & Roth, 2002), Trivia QA (Joshi et al., 2017b), SamSum (Gliwa et al., 2019), LCC (Guo et al., 2023), and RepoBench-P (Liu et al., 2023)).

**Table S.1:** Ablation study on the Gated DeltaNet block. Avg-PPL and Avg-Acc denote average perplexity and zero-shot commonsense reasoning accuracy (as in Table 3), respectively. All models have 400M parameters and are trained for 15B tokens on the same subset of FineWeb-Edu dataset (Penedo et al., 2024).

| Gated DeltaNet Ablations (400M) | Avg-PPL ($\downarrow$) | Avg-Acc ($\uparrow$) |
|---|---|---|
| Gated DeltaNet *w* Head Dim 128, | 27.35 | 47.26 |
| *Macro Design* | | |
| *w.* naive Delta Rule | 30.87 | 45.12 |
| *w/o.* Short Conv | 28.95 | 46.16 |
| *w/o.* Output Gate | 29.12 | 45.46 |
| *w/o.* Output Norm | 27.55 | 47.07 |
| *Normalization & Feature Map* | | |
| *w.* $L_1$-norm & ReLU | 30.79 | 45.92 |
| *w.* $L_1$-norm & 1+ELU | 30.34 | 46.05 |
| *w.* $L_1$-norm & SiLU | 30.18 | 46.09 |
| *w.* $L_2$-norm & ReLU | 27.67 | 46.94 |
| *w.* $L_2$-norm & 1+ELU | 27.58 | 47.17 |
| *Model Dimensions* | | |
| *w.* Head Dim 64 | 28.31 | 46.35 |
| *w.* Head Dim 256 | 27.13 | 47.38 |

## B.2 ABLATION STUDY

| Model | Wiki. ppl $\downarrow$ | LMB. ppl $\downarrow$ | LMB. acc $\uparrow$ | PIQA acc $\uparrow$ | Hella. acc_n $\uparrow$ | Wino. acc $\uparrow$ | ARC-e acc $\uparrow$ | ARC-c acc_n $\uparrow$ | SIQA acc $\uparrow$ | BoolQ acc $\uparrow$ | Avg. |
|---|---|---|---|---|---|---|---|---|---|---|---|
| *Hybrid Ablations (500M/15B)* | | | | | | | | | | | |
| Gated DeltaNet + SWA + Mamba2 | 24.02 | 28.20 | 34.77 | 67.08 | 40.84 | 50.74 | 60.35 | 28.83 | 38.94 | 61.49 | 47.88 |
| Gated DeltaNet + Mamba2 + SWA | 23.69 | 26.83 | 36.17 | 67.51 | 41.51 | 51.85 | 61.19 | 29.77 | 38.58 | 53.73 | 47.54 |
| Mamba2 + SWA + Gated DeltaNet | 24.14 | 25.21 | 36.79 | 64.96 | 41.18 | 52.01 | 60.90 | 30.03 | 38.07 | 59.44 | 47.92 |
| Mamba2 + Gated DeltaNet + SWA | **23.54** | **24.11** | 36.92 | 66.48 | 41.70 | 52.72 | 61.06 | 30.54 | 39.91 | 60.51 | **48.73** |

**Table S.2:** Ablation studies of Gated DeltaNet models. All evaluations are performed by using `lm-evaluation-harness` (Gao et al., 2021). All models use the Llama tokenizer and are trained on the same subset of the FineWeb-Edu dataset (Penedo et al., 2024).

Table S.1 presents ablation studies on the Gated DeltaNet block's components. Our experiments demonstrate that both the short convolution and output gate are crucial for model performance, while output normalization yields marginal improvements. Consistent with Yang et al. (2024b), we found L2 normalization to be essential for optimal performance, though the choice of feature map was less influential. Nevertheless, SiLU consistently outperformed other activation functions, aligning with observations from Qin et al. (2023a). Through empirical analysis, we determined that a head dimension of 128 provides an optimal trade-off between performance and computational efficiency. Additionally, Table S.2 demonstrates that among various hybrid architectures, the combination of Mamba2, Gated DeltaNet, and SWA in this specific order produces superior results.

