# OpenReview forum: "Gated Delta Networks: Improving Mamba2 with Delta Rule"
_ICLR.cc/2025/Conference — ICLR 2025 Poster_

### Official Review · Reviewer_3Thw · 2024-10-29

**Soundness:** 3
**Presentation:** 3
**Contribution:** 3
**Rating:** 8
**Confidence:** 5

**Summary:**

This paper proposes Gated DeltaNet, a novel model architecture combing gated recurrent mechanism and Delta update rule. Gated DeltaNet support chunk-parallel computation as efficiency optimization. Besides simple Gated DeltaNet, the paper also discuses the hybrid design with Sliding-Window Attention or Mamba layers.

In experiments, the paper shows strong modeling capability of Gated DeltaNet. First, Gated DeltaNet outperforms Transformer and other linear-complexity models in language modeling and zero-shot downstream tasks. Second, Gated DeltaNet shows a strong length-extrapolation ability in language modeling. In ablation studies, the paper makes comparisons with other design choices.

**Strengths:**

1. The paper is well written, showing the connection and improvement with previous works.

2. Gated DeltaNet makes kernel optimization to achieve better training and inference throughput, which is essential for downstream application and community reproduction.

3. Gated DeltaNet shows strong performance than other linear-complexity models, which is a good academic contribution in model architecture area.

**Weaknesses:**

1. This paper does not discuss the performance on long-context tasks. Since the quadratic complexity only matters when the sequence is long, long-context performance is the most important metric for linear-complexity models.

2. This paper does not compare Gated DeltaNet with other linear-complexity models on training and inference efficiency. Since Delta-rule is harder for parallel computation, there may be a trade-off between performance and computation cost.

**Questions:**

How do you think the application scenario of linear-complexity models? Since it is going to be common sense that linear complexity makes a sacrifice on long-context capability.

---

> ### Author Response · Authors · 2024-11-22
>
> Thank you for your positive review! As requested, we have provided long-context evaluation results in [General Response1](https://openreview.net/forum?id=r8H7xhYPwz&noteId=FwDZPReXwv) and training efficiency comparisons in [General Response2](https://openreview.net/forum?id=r8H7xhYPwz&noteId=JEEsR7mmqv).
>
> Regarding your question, we believe a more practical use case for linear-complexity models is their application as a fast "token mixer" in hybrid rnn-attention models or decoder-decoder architectures (Sun et al., 2024). Recent studies, such as those by Lieber et al. (2024) and Waleffe et al. (2024), have demonstrated that most attention layers (~90%) can be replaced by linear-complexity layers without sacrificing the performance.  In this work, we do find that hybridizing sliding window attention with gated delta rule outperforms hybridizing it with Mamba1, indicating that advancements in linear architectures continue to enhance hybrid models.
>
>
>
> References:
>
> - You Only Cache Once: Decoder-Decoder Architectures for Language Models, Sun et al, 2024, https://arxiv.org/abs/2405.05254
> - Jamba: A Hybrid Transformer-Mamba Language Model,  Lieber et al, 2024. https://arxiv.org/abs/2403.19887
> - An Empirical Study of Mamba-based Language Models, Waleffe et al, 2024. https://arxiv.org/abs/2406.07887

---

> > ### Comment · Reviewer_3Thw · 2024-11-22
> >
> > Thanks for your reply. I will maintain my positive score.

---

> > > ### Author Response · Authors · 2024-11-28
> > >
> > > Thank you for the positive score! We've revised the paper; please refer to [Summary of Paper Revisions](https://openreview.net/forum?id=r8H7xhYPwz&noteId=ayDvdRmYGW) for details.

---

### Official Review · Reviewer_jNuR · 2024-11-02

**Soundness:** 4
**Presentation:** 4
**Contribution:** 4
**Rating:** 6
**Confidence:** 4

**Summary:**

The paper introduces Gated DeltaNet as an enhancement to linear transformers, combining two memory management mechanisms: the gating mechanism from Mamba2 and the delta update rule from DeltaNet. Linear transformers, known for efficiency, often struggle with recall-intensive tasks. The gating mechanism enables flexible, rapid memory clearance, while the delta rule allows for targeted updates. By integrating these, Gated DeltaNet achieves both efficient memory erasure and selective memory retention, addressing limitations faced by linear transformers in handling extended sequences and complex memory recall tasks.

The proposed model demonstrates superior performance compared to Mamba2 and DeltaNet across various tasks, including language modeling, commonsense reasoning, and associative-recall. Additionally, the study explores hybrid models that combine Gated DeltaNet with sliding window attention or Mamba2 layers to further improve retrieval capacity. Extensive experiments validate the effectiveness of the gated delta rule, highlighting its advantage in managing memory over long sequences and maintaining hardware efficiency through chunkwise parallelism for training.

**Strengths:**

This paper gives detailed and structured introduction to a series of d^2 linear attention model e.g. GLA, DeltaNet and Mamba2, following a newly proposed Gated DeltaRule as its main contribution.  The improvements of delta rule can be considered as one strength in this paper, with a scalar-values decay term for easy training. Also, the UT transform for using Tensor Core with high-efficiency serves as a mitigation to the extended WY representation is unique, yet could be an individual interest to the community.

 The motivation for combining gating with the delta rule is compelling, showing how these two mechanisms address specific weaknesses in memory management. The writing in this paper is generally clear, structured, and detailed.

**Weaknesses:**

One weakness of the paper lies in the use of a scalar-value decay term, which, while intended to enhance memory management, does not consistently outperform Mamba2 in associative recall tasks. As seen in Table 2, although Gated DeltaNet shows a slight improvement in average accuracy across six associative recall tasks, Mamba2 still performs better in three of them. This raises questions about the efficacy of the decay term in improving recall performance. Intuitively, adding a decay term implies a rigid truncation of information over time, which may not necessarily benefit associative recall tasks that rely on longer memory retention. Providing case studies that track changes in hidden states during tasks could clarify the impact of the decay term on associative recall. Such insights would help to understand how the decay term influences memory retention and retrieval processes within the model, making the improvement more transparent and justifiable.

Another issue concerns the length extrapolation experiments involving DeltaNet. The paper would benefit from clearly labeling the four model types and specifying the experimental setup in detail. For example, it appears that DeltaNet performs even worse than Mamba in extrapolation, which contradicts findings from other experiments within the paper. This inconsistency suggests that additional clarification on the model configurations, including whether any of the models employed a hybrid architecture, is necessary. Given that attention mechanisms offer significant advantages in associative recall capabilities, it is essential to specify if the models were using sliding window attention or other hybrid elements, as this could substantially influence the results and better contextualize the model comparisons.

Typo:
- line 256, "As we can see in ." need a reference?
- line 313, 319, 334, "Training" -> "Training."
- line 374, "State size is strongly correlated with final performance." need a \paragraph{}.

**Questions:**

1. Is there any visualized results or case studies about why the introduction of scalar value decay term resulted in following improvements in downstream task like AR?

2. How about the training / inference speed e.g. throughput comparing with DeltaNet, Mamba2, and hybrid model e.g. Samba?

3. Does the UT transform really helps the acceleration of training? How about some ablation studies on w/ and w/o the UT transform?

---

> ### Author Response · Authors · 2024-11-22
>
> Thank you for your positive review!
>
> > One weakness of the paper lies in the use of a scalar-value decay term, which, while intended to enhance memory management, does not consistently outperform Mamba2 in associative recall tasks. As seen in Table 2, although Gated DeltaNet shows a slight improvement in average accuracy across six associative recall tasks, Mamba2 still performs better in three of them.
>
> Please see [General Response3](https://openreview.net/forum?id=r8H7xhYPwz&noteId=c4aRDdAAwQ). We observe a clear advantage of GatedDeltaNet over Mamba2.
>
> > This raises questions about the efficacy of the decay term in improving recall performance. Intuitively, adding a decay term implies a rigid truncation of information over time, which may not necessarily benefit associative recall tasks that rely on longer memory retention
>
> Yes, this is a valid point. As observed in [General Response3](https://openreview.net/forum?id=r8H7xhYPwz&noteId=c4aRDdAAwQ) NIAH-S1, the pass-key retrieval performance is hindered by the decay term, which affects memory retention. However, we did not observe a significant performance drop in the gated delta rule compared to the delta rule. This suggests that the decay term's influence is limited due to its **data-dependent** nature; theoretically, the model can learn to set these terms to 1 where necessary. In contrast, models with **data-independent** exponential decay face more pronounced issues, which is why the field has transitioned from data-independent approaches (e.g., S4, S5, RWKV4, LRU, RetNet) to data-dependent ones (e.g., HGRN, RWKV6, Mamba, GLA, GSA).
>
> We also want to emphasize that memory retrieval is influenced not only by memory retention but also by "memory collision" [1]. In complex contexts (e.g., NIAH-S2 and NIAH-S3), high memory retention can become a liability, as the model struggles to clear irrelevant information promptly. This leads to states that cannot effectively distinguish the original content, as they store a superposition of multiple pieces of information. This challenge was one of our primary motivations, and we believe the performance results in NIAH-S2 and NIAH-S3 convincingly support our argument.
>
> [1]  Linear Transformers Are Secretly Fast Weight Programmers, https://arxiv.org/abs/2102.11174
>
> > For example, it appears that DeltaNet performs even worse than Mamba in extrapolation, which contradicts findings from other experiments within the paper.
>
> Could you clarify this? We are not sure which experiments suggest that Mamba performs worse than DeltaNet in extrapolation.
>
> >  The paper would benefit from clearly labeling the four model types and specifying the experimental setup in detail. ........ Given that attention mechanisms offer significant advantages in associative recall capabilities, it is essential to specify if the models were using sliding window attention or other hybrid elements, as this could substantially influence the results and better contextualize the model comparisons.
>
> Thank you for the suggestion. In Figure 2, all models are pure 1.3B RNNs without sliding window attention, trained on 100B tokens. We will clarify this.
>
> > How about the training / inference speed e.g. throughput comparing with DeltaNet, Mamba2, and hybrid model e.g. Samba?
>
> Please see [General Response 2](https://openreview.net/forum?id=r8H7xhYPwz&noteId=JEEsR7mmqv)

---

> ### Comment · Reviewer_jNuR · 2024-11-22
>
> Thanks for the detailed explanations. I believe the experiments provided by the authors are very clear and address my concerns. I will maintain my original score to support acceptance.

---

> > ### Author Response · Authors · 2024-11-28
> >
> > Thank you for your positive score and we're glad to hear that your concerns have been addressed! Also, we've revised the paper; please refer to [Summary of Paper Revisions](https://openreview.net/forum?id=r8H7xhYPwz&noteId=ayDvdRmYGW) for details.

---

### Official Review · Reviewer_JW3m · 2024-11-04

**Soundness:** 3
**Presentation:** 3
**Contribution:** 3
**Rating:** 8
**Confidence:** 3

**Summary:**

This paper addresses the limitations of linear Transformers in handling long sequences and recall-intensive tasks. While linear Transformers reduce computational complexity compared to traditional Transformers, they often struggle with tasks that require remembering and recalling information over extended contexts. Authors focus on (1) Gating Mechanism: Allows the model to adaptively forget irrelevant information by applying a dynamic decay to the hidden state. This enables fast and efficient memory erasure but may not precisely target specific memories. (2) Delta Update Rule: Enables precise and targeted updates to memory by selectively replacing old key-value pairs with new ones, to propose a new approach called Gated Delta Rule, which combines the strengths of both mechanisms.

**Strengths:**

## Strengths

1. This paper propose a new linear Transformer that implements the gated delta rule, allowing for more effective memory management over long sequences.
2. The authors extend the delta rule's parallel algorithm to incorporate gating, ensuring hardware-efficient training using chunkwise parallelism and tensor core acceleration.
3. Experiments demonstrate that Gated DeltaNet can outperforms existing models like Mamba2 (which uses gating) and DeltaNet (which uses the delta rule) on various tasks.

**Weaknesses:**

## Weakness

1. The paper lacks an in-depth theoretical analysis of why the combination of gating and the delta rule improves memory capacity and how it affects the model's ability to handle long sequences, providing theoretical insights regarding the memory retention and forgetting properties of GatedDeltaNet would strengthen the paper's contributions.
2. Although the authors claims that Gated DeltaNet generalizes better to longer sequences than DeltaNet / Mamba2, the experimental results on length extrapolation are limited (only one analysis on PG19). The evaluation focuses on sequences up to a certain length without exploring the model's performance on substantially longer sequences. Expanding the extrapolation experiments to include much longer sequences would provide stronger evidence.

**Questions:**

1. Could you provide quantitative metrics on the computational efficiency of Gated DeltaNet compared to other models, such as training/inference speed and memory consumption?

I will read the rebuttal in the discussion period and adjust my evaluation.

---

> ### Author Response · Authors · 2024-11-25
>
> We thank the reviewer for their feedback.
>
> For weakness1,  we provide a more formal treatment of the gated delta rule through the lens of the online learning framework proposed by Longhorn (https://arxiv.org/abs/2407.14207), where recurrent state updates are formulated as solutions to an online learning problem with objective function $L_t(S)$.
>
> Table 1 summarizes recent RNN models' online learning objective and update, typically including a penalty term to prevent the model state from diverging too far from its previous or decayed value. Linear Attention (LA) and Mamba2 use a simple linear prediction loss $\langle S_t k_t, v_t\rangle$, while Longhorn uses a more expressive online regression objective $|S_tk_t - v_t|^2$ for better modeling of key-value associations. The resulting Longhorn's update rule closely resembles the delta update rule, suggesting the superiority of (gated) delta rule over Mamba2 in in-context associative recall.
>
> Finally, both Mamba2 and Gated DeltaNet relax the first term by introducing an adaptive scaling factor $\alpha_t$, allowing $S_t$ to deviate from $S_{t-1}$ in a controlled manner. This modification enables dynamic memory clearance. Our empirical results in [General Response 3](https://openreview.net/forum?id=r8H7xhYPwz&noteId=c4aRDdAAwQ) support this theoretical insight.
>
> | Method | Online Learning Objective $L_t(S)$ | Online Update |
> |--------|------------------------------------------------|---------------|
> | LA | $\|S_t - S_{t-1}\|_F^2 - 2\langle S_t k_t, v_t\rangle$ | $S_t = S_{t-1} + v_t k_t^T$ |
> | Mamba2 | $\|S_t - \alpha_t S_{t-1}\|_F^2 - 2\langle S_t k_t, v_t\rangle$ | $S_t = \alpha_t S_{t-1} + v_t k_t^T$ |
> | Longhorn | $\|S_t - S_{t-1}\|_F^2 - \beta_t \|S_t k_t - v_t \|^2$ | $S_t = S_{t-1}(I - \epsilon k_t k_t^T) + \epsilon_t v_t k_t^T, \epsilon_t=\frac{\beta_t}{1+\beta_tk_t^\top k_t}$ |
> | DeltaNet | $\|{S_t - S_{t-1}}\|^2_F - 2<S_t k_t, \beta_t(v_t - S_{t-1}k_t)>$ | $S_t = S_{t-1}(I - \beta_t k_t k_t^\top) + \beta_t v_t k_t^\top$ |
> | Gated DeltaNet | $\|S_t - \alpha_t S_{t-1}\|_F^2 - 2<S_t k_t, \beta_t(v_t- \alpha_tS_{t-1}k_t)>$ | $S_t = S_{t-1}(\alpha_t(I - \beta_t k_t k_t^\top)) + \beta_t v_t k_t^\top$ |
>
> **Table 1:** Comparison of different linear RNN models and their corresponding online learning objectives using the framework from [Longhorn]. For convenience, we simplify Longhorn's vector-valued $\beta$ to scalar $\beta$. DeltaNet can also be interpreted as optimizing Longhorn's objective through one-step explicit gradient descent, rather than seeking the global optimum.

---

> > ### Author Response · Authors · 2024-11-25
> >
> > >  the experimental results on length extrapolation are limited (only one analysis on PG19).
> >
> > We plan to expand our evaluation to diverse benchmarks across domains. Here are our initial results on the CodeParrot benchmark, with more comprehensive evaluations forthcoming.
> >
> > | Model | 2K | 4K | 6K | 8K | 10K | 12K | 14K | 16K | 18K | 20K |
> > |-------|-----|-----|-----|-----|-----|-----|-----|-----|-----|-----|
> > | Mamba1 | 19.26 | 15.92 | 15.46 | 15.12 | 14.64 | 14.27 | 13.69 | 13.12 | 12.71 | 12.19 |
> > | Mamba2 | 12.76 | 9.31 | 9.09 | 9.12 | 9.04 | 8.91 | 8.74 | 8.52 | 8.27 | 8.03 |
> > | GatedDeltaNet | 11.82 | 8.35 | 7.98 | 7.86 | 7.72 | 7.57 | 7.36 | 7.15 | 6.94 | 6.75 |
> > | Samba | 11.85 | 7.56 | 7.30 | 7.19 | 7.06 | 6.92 | 6.70 | 6.47 | 6.30 | 6.07 |
> > | GatedDeltaNet-H2 | 10.99 | 7.25 | 7.03 | 6.98 | 6.88 | 6.74 | 6.57 | 6.39 | 6.24 | 6.03 |
> > | GatedDeltaNet-H1 | 10.64 | 6.92 | 6.69 | 6.66 | 6.53 | 6.40 | 6.23 | 6.05 | 5.91 | 5.75 |
> >
> > > Expanding the extrapolation experiments to include much longer sequences would provide stronger evidence.
> >
> > Thank you for your suggestion. However, on a single 80G GPU we cannot further increase the sequence length. While it is possible to apply "state passing" (of RNN state or SWA's sliding window KV cache) for evaluation, our current inference code is unfortunately primitive and requires additional engineering efforts, which could not be done in such a short period of time. We plan to evaluate on longer sequences in the future.
> >
> > > Could you provide quantitative metrics on the computational efficiency of Gated DeltaNet compared to other models, such as training/inference speed and memory consumption?
> >
> > Please see [GeneralResponse2](https://openreview.net/forum?id=r8H7xhYPwz&noteId=JEEsR7mmqv)

---

> > > ### Comment · Reviewer_JW3m · 2024-11-27
> > > **Thanks for your rebuttal**
> > >
> > > Thanks a lot for your response. After reading your rebuttal, I think most of my concerns have been addressed. So, I decide to change my rating from 6 -> 8 :)

---

> > > > ### Author Response · Authors · 2024-11-28
> > > >
> > > > Thank you for raising the score! We're glad to hear that most of your concerns have been addressed. We've revised the paper; please refer to [Summary of Paper Revisions](https://openreview.net/forum?id=r8H7xhYPwz&noteId=ayDvdRmYGW) for details.

---

### Official Review · Reviewer_eAAH · 2024-11-08

**Soundness:** 2
**Presentation:** 3
**Contribution:** 2
**Rating:** 6
**Confidence:** 2

**Summary:**

To mitigate quadratic scaling with sequence length of standard Transformers, Linear Transformers with kernelized dot-product linear attention are proposed. In this paper, the authors propose Gated DeltaNet, which leverages the gating mechanism from Mamba2 and the delta update rule from DeltaNet. The hybrid architecture of Gated DeltaNet, SWA, Mamba2 layers outperforms baselines in language modeling, reasoning, and recall-intensive tasks.

**Strengths:**

- This paper is well-organized and clearly written.
- Their presentation, especially highlighting contribution points, was good to understand and follow.
- The proposed method is simple, but seems effective in various tasks.

**Weaknesses:**

- How are the baselines trained? Those few-shot performance look a little lower than other papers (like Mamba). And also, could you compare with some other well-pretrained Transformer models to show the effectiveness of the proposed architecture?
- State sizes of Gated DeltaNet H1 and H2 seem much larger than all the other baselines---models that use SWA have the larger sizes. In case of Gated DeltaNet (with 256 state size) performance looks similar to Mamba2 and DeltaNet, where the gated delta rule can be seen as having minimal impact on performance. Also, as Samba already shows a very good performance, I guess the performance gain would come from hybrid structure, not from gated delta rule itself.
- I suggested the authors to compare efficiency of models, which is an important factor for a new architecture.
- In length extrapolation experiments, why does Gated DeltaNet show robust performance while Mamba2 and DeltaNet show higher perplexity as the length grows. Gated DeltaNet is the combination of both methods.

**Questions:**

- There are some typos in Section 3.1 (e.g., L.239 or L.256).

---

> ### Author Response · Authors · 2024-11-22
>
> Thanks for your review!
>
> ## Weakness 1
> > How are the baselines trained?
>
> All models are trained under the same settings, leveraging the codebases of [TinyLLaMa](https://github.com/jzhang38/TinyLlama) and [FlashLinearAttention](https://github.com/sustcsonglin/flash-linear-attention).
>
> > Those few-shot performance look a little lower than other papers (like Mamba)
>
> The observed performance gap arises from differences in training settings. For example, Mamba1-2 was trained on 300B tokens, whereas due to computational constraints, we trained our models on only 100B tokens. This discrepancy primarily accounts for the performance difference. When compared to GLA [1], which also trained 1.3B models on 100B tokens, our results outperform theirs, suggesting our numbers are reasonable within this context.
>
> [1] Gated Linear Attention Transformers with Hardware-Efficient Training. https://arxiv.org/abs/2312.06635
>
>
> ## Weakness2
> >In case of Gated DeltaNet (with 256 state size) performance looks similar to Mamba2 and DeltaNet, where the gated delta rule can be seen as having minimal impact on performance
>
> Please see [GeneralResponse3](https://openreview.net/forum?id=r8H7xhYPwz&noteId=c4aRDdAAwQ).
>
> >  I guess the performance gain would come from hybrid structure, not from gated delta rule itself.
>
> We agree that the performance gain in retrieval tasks primarily stems from attention layers, as is well-documented in linear RNN literature. However, we believe that a more expressive recurrent state update rule, such as the gated delta rule, contributes to overall performance gains in other tasks. For example, as shown in [GeneralResponse1](https://openreview.net/forum?id=r8H7xhYPwz&noteId=FwDZPReXwv), GatedDeltaNet-H2 (i.e., GatedDeltaNet + SWA) outperforms Samba (i.e., Mamba1 + SWA), indicating that the linear recurrent layer plays a significant role in long-context understanding.
>
> ## Weakness3
> As requested, the training efficiency comparison is available in [GeneralResponse2](https://openreview.net/forum?id=r8H7xhYPwz&noteId=JEEsR7mmqv).
>
> ## Weakness4
>
> DeltaNet lacks a memory-clearing mechanism, which leads to a rapid increase in perplexity as sequence length grows. In contrast, both Mamba2 and GatedDeltaNet incorporate gating mechanisms that are highly effective in mitigating this issue.
>
> We hypothesize that the rise in perplexity for Mamba2 with increasing sequence length may be due to its memory capacity becoming saturated. When this happens, Mamba2 employs a more aggressive memory-clearing mechanism, resulting in the loss of additional past information and, consequently, higher perplexity. On the other hand, GatedDeltaNet benefits from the delta rule, which enables adaptive and targeted memory updates. This design allows GatedDeltaNet to avoid the need for such aggressive memory clearance, maintaining robust performance even with longer sequences.

---

> ### Comment · Reviewer_eAAH · 2024-11-25
>
> Sorry for the late reply, and thanks for the detailed responses, as well as the additional experimental results in general responses!
>
> To be honest, I'm not the expert in this field, so it was quire challenging to gauge the novelty of this work and how effectively this new structure can address the limitations in previous work.
> I raised my score to 6, and I'm inclined to agree with the other reviewer's positive opinions.

---

> ### Author Response · Authors · 2024-11-25
>
> Thank you for raising the score!  We've revised the paper; please refer to [Summary of Paper Revisions](https://openreview.net/forum?id=r8H7xhYPwz&noteId=ayDvdRmYGW) for details.

---

### Author Response · Authors · 2024-11-22
**General Response3:  Recall-Intensive Task Performance**

Our empirical results demonstrate that Gated DeltaNet **consistently** outperforms Mamba2 (in average) across model scales **when controlling for state size**, though the margin of improvement is modest. The relatively small performance gap may be attributed to the predominant error type - repetition - as documented in Appendix E of the (https://arxiv.org/pdf/2407.05483). This error pattern is characteristic of instruction-unaligned smaller models regardless of their update mechanisms, suggesting that raw performance metrics may not fully capture the architectural advantages of different update rules.

To rigorously evaluate the retrieval capabilities of gated delta rule compared to its standalone counterparts, we conducted controlled experiments using the increasingly challenging Needle-In-A-Haystack Single (NIAH-S) benchmark suite from RULER (COLM'24), progressing from NIAH-1 through NIAH-3. The results below quantify these comparative advantages (all evaluated models are 1.3B pure RNN models)

### S-NIAH-1 (aka pass-key retrieval)
| Model           | 1K   | 2K   | 4K   | 8K   |
|------------------|-------|-------|-------|-------|
| DeltaNet         | 97.4  | 96.8  | 99.0  | 98.8  |
| GatedDeltaNet    | 98.4  | 88.4  | 91.4  | 91.8  |
| Mamba2           | 99.2  | 98.8  | 65.4  | 30.4  |

### S-NIAH-2 (aka needle (number) in a haystack)
| Model           | 1K   | 2K   | 4K   | 8K   |
|------------------|-------|-------|-------|-------|
| DeltaNet         | 98.4  | 45.6  | 18.6  | 14.4  |
| GatedDeltaNet    | 100   | 99.8  | 92.2  | 29.6  |
| Mamba2           | 99.4  | 98.8  | 56.2  | 17.0  |

### S-NIAH-3 (aka needle (word) in a haystack)
| Model           | 1K   | 2K   | 4K   |
|------------------|-------|-------|-------|
| DeltaNet         | 85.2  | 47.0  | 22.4  |
| GatedDeltaNet    | 86.6  | 84.2  | 27.6  |
| Mamba2           | 64.4  | 47.6  | 4.6  |

The S-NIAH benchmark suite reveals distinct performance patterns across architectures. In S-NIAH-1 (pass-key retrieval), DeltaNet excels due to its pure delta update mechanism, with GatedDeltaNet showing competitive performance despite minor degradation from the proposed gating mechanism. Notably, Mamba2's performance deteriorates sharply beyond 2K sequence lengths.

Unlike S-NIAH-1, S-NIAH-2/3 grounds the models in realistic text data, which is significantly more complex and contains noisier signals. DeltaNet performs well with short sequences (1K) but struggles with longer sequences due to the absence of a memory-erasing mechanism. On the other hand, GatedDeltaNet maintains robust performance up to 4K tokens in S-NIAH-2 (92.2%) and shows significant advantages in S-NIAH-3, outperforming both baselines. These results empirically validate our theoretical motivation for incorporating a memory erasing mechanism, particularly evident in tasks requiring selective information retention. Mamba2's performance degrades more rapidly with increased sequence length and task complexity, dropping to 4.6% in S-NIAH-3 at 4K tokens compared to GatedDeltaNet's 27.6%. The progressive difficulty across the benchmark suite effectively demonstrates the complementary benefits of combining gating with delta updates, particularly for complex information retrieval scenarios.

---

### Author Response · Authors · 2024-11-22
**General Response2:  Efficiency Benchmarking**

We measure the running speed on a single H100 machine, varying the sequence length and batch size while keeping the total number of tokens fixed at 32K.  The training throughputs (tokens per second) are summarized below.

| Model                | 2K x 16 | 4K x 4 | 8K x 2 | 16K x 2 | 32K x 1 |
|-----------------------|---------|--------|--------|---------|---------|
| Transformer          | 55.0K   | 47.6K  | 37.9K  | 26.5K   | 16.6K   |
| DeltaNet             | 45.9K   | 45.6K  | 45.6K  | 45.2K   | 43.0K   |
| GatedDeltaNet (headdim256)       | 46.1K   | 45.7K  | 45.8K  | 45.9K   | 41.7K   |
| Mamba1               | 38.2K   | 38.1K  | 37.8K  | 37.6K   | 41.7K   |
| Mamba2               | 48.1K   | 48.2K  | 48.2K  | 48.1K   | 48.2K   |
| GatedDeltaNet-H2 (GatedDeltaNet+SWA) headdim256      | 55.6K   | 53.4K  | 52.5K  | 52.5K   | 48.9K   |
| Samba (Mamba+SWA)    | 45.0K   | 44.0K  | 43.3K  | 42.8K   | 45.5K   |

As mentioned in the paper, the gated delta rule introduces only marginal efficiency overhead compared to the original delta rule operation while performing competitively with alternatives such as Mamba-2.

Our analysis of inference efficiency is currently limited by Hugging Face's API performance, which is much slower (see https://github.com/state-spaces/mamba/issues/578 for a discussion). As such, we were unable to compare the inference speed against the official Mamba impelmentation.  We plan to implement all models in a faster framework such as VLLM for a more equitable comparison. Based on comparable training throughput and state sizes, we expect Gated DeltaNet's inference speed to match Mamba2. This is because both prefilling time should be roughly equivalent (indicated by the training time), and recurrent inference speed is primarily determined by state size in memory-bounded operations.

---

### Author Response · Authors · 2024-11-22
**General Response1:  Long Context Evaluation**

We evaluate all 1.3B models on 14 tasks from the LongBench benchmark (ACL '24).


| Model              | NarrativeQA | QasperQA | MFQA  | HotpotQA | 2WikiMQA | Musique | GovReport | QMSum | MultiNews | TRec  | TriviaQA | SamSum | LCC   | RepoBench-P | Average |
|--------------------|-------------|----------|-------|----------|----------|---------|-----------|-------|-----------|-------|----------|--------|-------|-------------|---------|
| Transformer++      | 11.83       | 9.25     | 10.00 | 10.87    | 4.21     | 6.14    | 7.42      | 15.83 | 6.62      | 16.93 | 13.48    | 3.93   | 17.18 | 18.69       | 10.88   |
| RetNet             | 12.10       | 10.71    | 19.13 | 10.67    | 18.05    | 5.80    | 4.83      | 15.76 | 7.85      | 19.00 | 18.05    | 12.77  | 14.15 | 17.93       | 13.34   |
| HGRN2              | 10.67       | 12.10    | 19.05 | 11.29    | 15.73    | 6.00    | 5.24      | 15.10 | 9.19      | 16.00 | 15.76    | 10.33  | 18.64 | 20.76       | 13.28   |
| Mamba              | 13.39       | 10.08    | 20.38 | 10.10    | 16.66    | 6.04    | 7.21      | 15.94 | 8.39      | 23.12 | 21.93    | 11.17  | 17.93 | 19.02       | 14.38   |
| DeltaNet           | 12.90       | 10.75    | 21.47 | 10.92    | 13.20    | 5.06    | 6.48      | 13.52 | 7.22      | 15.50 | 23.33    | 11.60  | 17.59 | 20.28       | 13.56   |
| Mamba-2            | 11.14       | 11.33    | 18.63 | 11.75    | 15.09    | 6.74    | 6.67      | 14.49 | 7.35      | 13.00 | 23.60    | 8.36   | 17.87 | 20.63       | 13.33  |
| **Gated DeltaNet** | 14.11       | 13.97    | 23.31 | 13.72    | 14.39    | 5.79    | 7.52      | 16.41 | 7.89      | 29.97 | 22.43    | 22.98  | 18.72 | 22.05       | 16.66   |
| Samba (Mamba+SWA)            | 12.51       | 12.85    | 25.38 | 11.23    | 19.68    | 6.82    | 9.98      | 15.70 | 10.96     | 20.00 | 22.14    | 22.84  | 18.08 | 21.05       | 16.37   |
| **Gated DeltaNet-H2** | 12.65   | 12.97    | 27.08 | 12.67    | 20.59    | 7.46    | 10.42     | 16.19 | 12.98     | 40.50 | 22.71    | 27.89  | 19.93 | 19.18       | 18.80   |
| **Gated DeltaNet-H1** | 14.49   | 12.32    | 26.64 | 12.55    | 23.59    | 6.12    | 9.14      | 16.06 | 12.76     | 33.50 | 23.90    | 26.75  | 15.50 | 19.18       | 18.03   |

We can see GatedDeltaNet clearly outperform other sub-quadratic models.  It is noteworthy that Transformer++ perform poorly on certain tasks such as SamSum, 2WikiMQA due to its incapability to extrapolate effectively beyond its training sequence length.


Reference:
- LongBench: A Bilingual, Multitask Benchmark for Long Context Understanding https://arxiv.org/abs/2308.14508

---

### Author Response · Authors · 2024-11-28
**Summary of Paper Revisions**

Dear Reviewers,

We have uploaded a revised version of our paper with major changes highlighted in red.  We have also restructured the paper, moving the small-scale language model experiments to the appendix to accommodate new experimental results. Here are the key updates:

1. In Section 3.1, we now motivate the gated delta rule from an online learning perspective, under Reviewer JW3m's suggestion.

2. In Section 4, we have substantially expanded our evaluations:

   - Added challenging Single Needle In a Haystack (S-NIAH) synthetic dataset evaluation to address concerns from Reviewers eAAH and jNuR regarding the marginal improvements of Gated DeltaNet over Mamba2 in real-world in-context retrieval task.

   - Extended the length extrapolation evaluation to cover 6 tasks, strengthening our claims and responding to Reviewer JW3m's feedback

   - Included training throughput efficiency benchmarks to address all reviewers' concerns about computational efficiency.

   - Added comprehensive long-context evaluation results on Longbench, addressing Reviewer 3Thw's concern

We believe the more comprehensive evaluation, speed benchmarking, and extended discussion on delta rule, suggested by reviewers, strengthen the paper a lot.  Thank you for your constructive feedback.

Best regards,

Authors

---

### Meta-Review · Area_Chair_Yg5T · 2024-12-18

**Metareview:**

This paper introduces a new sequence model with linear scaling relative to the input length. Namely, authors combined the gating mechanism from Mamba-2 and DeltaNet's delta rule to get the benefits of both: gating enables rapid memory erasure and the delta rule facilitates targeted updates. Leveraging their proposal, authors introduced model block architectures that combine their Gated Delta operation with Mamba-2 layers and sliding window attention. Stacks of such blocks are somewhat extensively evaluated against a number of models comprising RNNs, quadratic transformers, and hybrid models.

The idea is simple and well justified. Evaluations are carried out on a number of different settings and show the proposed models to perform well both in terms of language modeling and reasoning, but also in terms of retrieval in needle-in-a-haystack settings. Training throughput is also observed to be higher than a number of alternatives.

I would note however that the evaluation is limited in some ways. For instance, it's unclear to what extent performance scales with training compute budget for this architecture since the authors focus only on a 1.3B parameters/100B tokens setup. Moreover, all comparisons are made against models trained by the authors. Running the setup proposed independently and comparing against numbers obtained by others would increase the confidence in the presented evidence. I also think the paper lacks evaluations of inference time performance and it would be great to see a comparison of test time throughput and memory footprint against transformers under KV-caching for instance.

Despite the limitations mentioned above, I would say the presented evidence suffices to indicate that the proposed architecture may improve upon the state-of-the-art and thus would be of interest to the community.

**Additional Comments On Reviewer Discussion:**

Reviews mostly asked for expansion of the evaluations and a more formal justification of the interventions in the architecture and their effect. The authors added new evals (e.g., retrieval and throughput) and also included a new section with an analysis of online learning objectives induced by various architectures, which greatly improved the manuscript.

---

### Decision · Program_Chairs · 2025-01-22

Accept (Poster)